# Engineering of a fluorescent chemogenetic reporter with tunable color for advanced live-cell imaging

Hela Benaissa[1,2], Karim Ounoughi[2], Isabelle Aujard[2], Evelyne Fischer[3], Rosette Goïame[3], Julie Nguyen [4], Alison G. Tebo [1,2,9], Chenge Li[2,10,11], Thomas Le Saux[2], Giulia Bertolin [5], Marc Tramier [5], Lydia Danglot [4,6], Nicolas Pietrancosta[1,7], Xavier Morin [3], Ludovic Jullien [2] & Arnaud Gautier [1,2,8 ✉]

Biocompatible fluorescent reporters with spectral properties spanning the entire visible spectrum are indispensable tools for imaging the biochemistry of living cells and organisms in real time. Here, we report the engineering of a fluorescent chemogenetic reporter with tunable optical and spectral properties. A collection of fluorogenic chromophores with various electronic properties enables to generate bimolecular fluorescent assemblies that cover the visible spectrum from blue to red using a single protein tag engineered and optimized by directed evolution and rational design. The ability to tune the fluorescence color and properties through simple molecular modulation provides a broad experimental versatility for imaging proteins in live cells, including neurons, and in multicellular organisms, and opens avenues for optimizing Förster resonance energy transfer (FRET) biosensors in live cells. The ability to tune the spectral properties and fluorescence performance enables furthermore to match the specifications and requirements of advanced super-resolution imaging techniques.

[1] Sorbonne Université, École Normale Supérieure, Université PSL, CNRS, Laboratoire des Biomolécules, LBM, 75005 Paris, France. [2] PASTEUR, Department of Chemistry, École Normale Supérieure, Université PSL, Sorbonne Université, CNRS, 75005 Paris, France. [3] Institut de Biologie de l'ENS (IBENS), École Normale Supérieure, CNRS, INSERM, Université PSL, 75005 Paris, France. [4] Université de Paris, NeurImag Imaging Facility, Institute of Psychiatry and Neuroscience of Paris, INSERM U1266, 75014 Paris, France. [5] University of Rennes, Centre National de la Recherche Scientifique (CNRS), (IGDR) Institute of Genetics and Development of Rennes, Unité Mixte de Recherche (UMR) 6290, F-35000 Rennes, France. [6] Université de Paris, Institute of Psychiatry and Neuroscience of Paris, INSERM U1266, Membrane Traffic in Healthy & Diseased Brain, 75014 Paris, France. [7] Neuroscience Paris Seine-Institut de Biologie Paris Seine (NPS-IBPS) INSERM, CNRS, Sorbonne Université, Paris, France. [8] Institut Universitaire de France, Paris, France. [9] Present address: Janelia Research Campus, Howard Hughes Medical Institute, Ashburn, VA 20147, USA. [10] Present address: Department of Obstetrics and Gynecology, Ren Ji Hospital, School of Medicine, Shanghai Jiao Tong University, Shanghai, China. [11] Present address: State Key Laboratory of Oncogenes and Related Genes, Shanghai Cancer Institute, Ren Ji Hospital, School of Medicine, Shanghai Jiao Tong University, Shanghai, China. ✉email: arnaud.gautier@sorbonne-universite.fr

Fluorescent labels and biosensors have revolutionized the way we study living systems. Targeted to specific biomolecules or cells, they allow non-invasive imaging of the machinery that govern cells and organisms in real time. The discovery and development of fluorescent proteins (FPs) have been a transformative milestone in molecular and cell imaging[1]. Genetic fusions to FPs allowed cell biologists to visualize the dynamics of proteins in living cells and organisms with unprecedented spatial and temporal resolution. The multiplicity of optical properties of FPs allowed multicolor imaging and the development of various biosensors. Recently, fluorescent chemogenetic reporters consisting in synthetic organic dyes anchored covalently or noncovalently to genetically encoded protein tags[2–6] have challenged the conventional paradigm of fluorescent proteins and opened new prospects for innovative on-demand bioimaging and biosensing[7–9]. These hybrid systems combine the targeting selectivity of genetic tags with the advantages of synthetic organic fluorophores, which can be very bright and photostable, and provide spectral properties covering most of the visible spectrum[10–12].

Here, we present the engineering of a fluorescent chemogenetic reporter with highly tunable spectral and optical properties. Spectral tuning was achieved by extensive modulation of the electronic properties of the embedded chromophore in the Fluorescence-Activating and absorption-Shifting Tag (FAST). Prototypical FAST is a small protein tag (14 kDa) with reduced genetic footprint, which was originally designed to fluoresce instantaneously upon non-covalent binding of fluorogenic 4-hydroxybenzylidene rhodanine (HBR) derivatives that are inherently non-fluorescent unless bound to FAST[13]. Recent studies showed that FAST enabled (i) to circumvent some limitations of conventional fluorescent proteins—allowing for instance the imaging of proteins in strict anaerobic organisms[14,15] or the visualization of protein secretion by bacteria in real time[16,17]—and (ii) to bioengineer functional biosensors for monitoring intracellular analytes[18], and observing protein–protein interactions with high spatial and temporal resolution[19]. Capable of binding various membrane permeant and membrane-impermeant HBR derivatives[13,20,21], prototypical FAST exhibited great potential for engineering a multifunctional fluorescent chemogenetic reporter with tunable color through chromophore modulation. The challenge lied in obtaining a rainbow of color from a single protein tag, as spectral tuning necessitated modulation of the electronic properties of the embedded chromophore, and thus structural change. We achieved this engineering feat by replacing the rhodanine found in HBR derivatives by isosteric heterocycles exhibiting various electron-withdrawing abilities in order to obtain emission wavelengths that cover the visible spectrum from blue ($\lambda_{em}$ 473 nm) to red ($\lambda_{em}$ 616 nm) (Fig. 1a–c). Cycles of directed protein evolution allowed the generation of a promiscuous FAST variant—named pFAST—with optimized chromophore binding affinity and fluorescence brightness. The ability to tune the fluorescence color from blue to red by choosing a different live-cell compatible fluorogenic chromophore provides a great experimental versatility for multicolor live-cell imaging. We demonstrate that this tunable fluorescent chemogenetic reporter allows investigators to face a large variety of experimental scenarios, provides interesting possibilities for optimizing and quantifying FRET biosensors in live cells, and can have spectroscopic attributes suitable for the most advanced imaging techniques, including STED super-resolution microscopy.

## Results

### Molecular spectral tuning.
Extensive spectral tuning necessitated to find a range of fluorogenic chromophores that displayed a wide variety of electronic properties and could still be recognized by a single binding pocket. To keep structural changes as minimal as possible, we replaced the electron-withdrawing rhodanine (R) heterocycle found in the push-pull fluorogenic HBR derivatives with isosteric pseudothiohydantoin (P), 2,4-thiazolidinedione (T) and 2,4-oxazolidinedione (O). Finer tuning was achieved by secondly varying the substituents on the electron-donating phenol, as previously achieved within the HBR series[21] (See Fig. 1a for structures). Just like HBR analogs[13], HBX ($X$ = P, T, or O) compounds are mainly protonated at physiological pH (their p$K_A$s range from 8.15 to 9.05) (Supplementary Fig. 1), they undergo a significant absorption red-shift upon deprotonation (Supplementary Fig. 1), and they are inherently non-fluorescent in solution. Preliminary screening showed that FAST was able to bind all chromophores, albeit with low to modest affinity (Supplementary Table 1). Like HBR analogs, the HBX ($X$ = P, T, or O) analogs displayed red-shifted absorption when bound to FAST at physiological pH (Supplementary Table 1), suggesting that the binding pocket of FAST stabilizes their deprotonated phenolate state in a similar fashion[13]. The corresponding bimolecular assemblies were all fluorescent, although in average dimmer than those formed with the HBR analogs, with absorption/emission peaks blue-shifted by ~35/40 nm (for the HBP analogs), ~65/65 nm (for the HBT analogs), and ~85/75 nm (for the HBO analogs) relative to their HBR-based counterpart. In a given series, changing the number and/or nature of the substituents present on the phenol moiety allowed us furthermore to finely tune the absorption/emission peaks of the FAST assemblies (Fig. 1, Supplementary Table 1).

### Optimization by directed protein evolution.
To obtain an optimized fluorescent chemogenetic reporter with tunable color, we engineered a FAST variant with extended chromophore binding promiscuity. Because the chromophores of the HBR, HBP, HBT, and HBO series were isosteric, we hypothesized that mutations able to increase the binding affinity and brightness for one chromophore could be sufficient to generate a promiscuous variant displaying improved properties with all chromophores. Directed protein evolution coupled with rational design have previously been an efficient method to generate FAST variants forming bright assembly with new chromophore[22] or having orthogonal chromophore specificity[23]. We thus used a combinatorial library of variants generated by random mutagenesis and displayed on yeast cells. Iterative rounds of fluorescence-activated cell sorting (FACS) decreasing progressively chromophore concentrations through rounds allowed us to identify variants forming tighter and brighter assemblies with chromophores of the HBX series (Fig. 2a–d, Supplementary Note 1, Supplementary Figs. 2–4, Supplementary Tables 2–9). Beneficial mutations were combined to generate oFAST, a variant with the mutations Q41L, D71N, V83I, M109L and S117R, which forms a blue fluorescent assembly with HBO-3,5DM 11-fold tighter and 3-fold brighter than FAST:HBO-3,5DM (Fig. 2b, Supplementary Table 2). In parallel, the genes of five variants displaying improved properties with HBP-3,5DM were recombined by DNA shuffling to generate a new library (library B, Fig. 2a). FACS screening allowed us to identify (i) tFAST, a variant with the mutations G25R, Q41K, S72T, A84S, M95A, M109L, and S117R relative to FAST, which forms a cyan fluorescent assembly with HBT-3,5DM 5-fold tighter and 1.5-fold brighter than FAST:HBT-3,5DM (Fig. 2c, Supplementary Table 3), and (ii) an improved variant for HBP-3,5DM bearing the mutations K17N, G21E, G25R, A30V, Q41L, S72T, V83A, M95T, S117R, which gave, after addition of the mutation M109L found in several other selected variants, the variant pFAST able to form a green fluorescent assembly with HBP-3,5DM 25-fold tighter and 2.5-fold brighter than FAST:HBP-3,5DM (Fig. 2d, Supplementary Table 4).

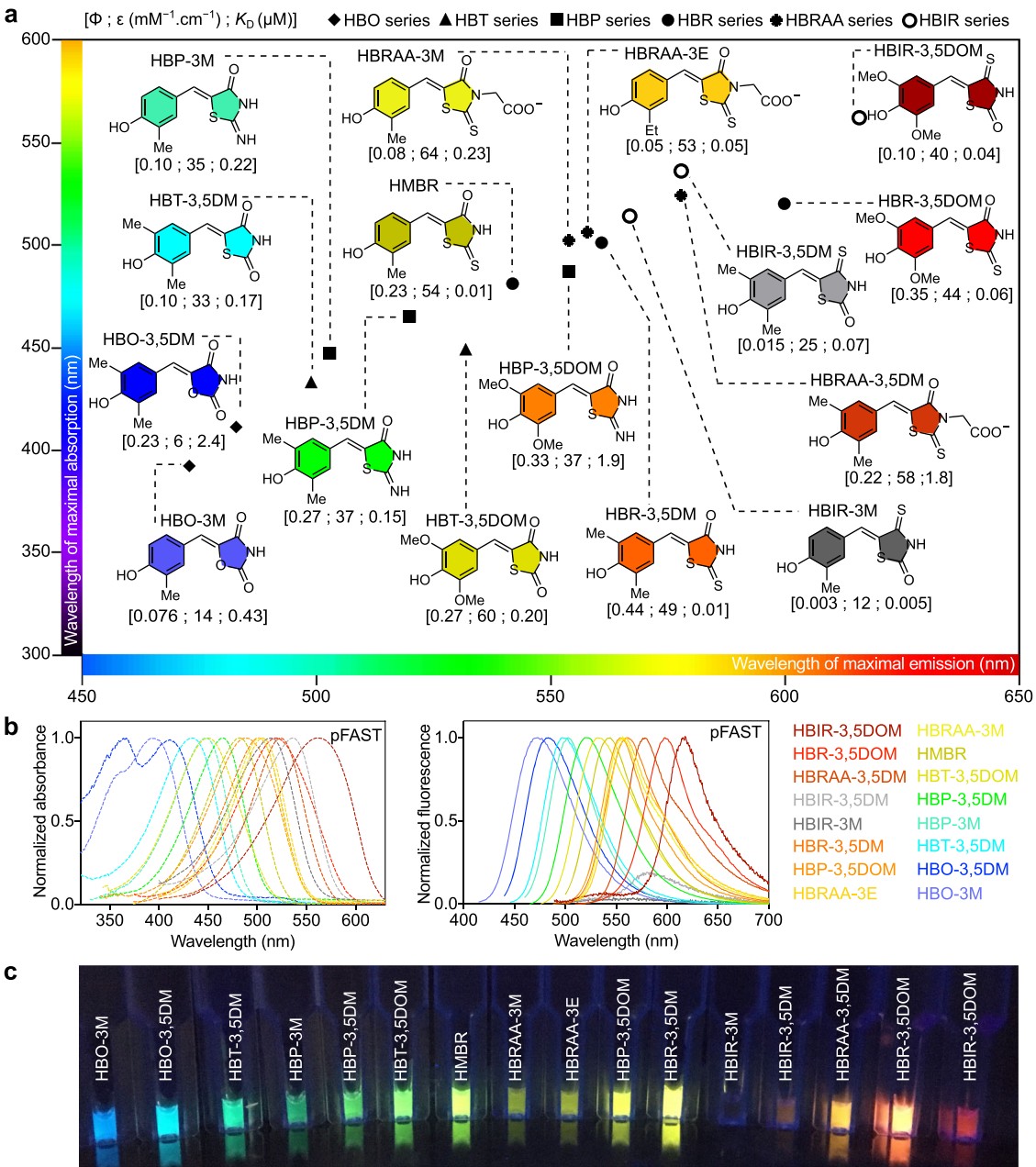

**Fig. 1 The engineered protein tag pFAST is a fluorescent chemogenetic reporter with tunable color. a** Structures of the chromophores of the HBO, HBT, HBP, HBR, HBIR, and HBRAA series positioned according to the wavelengths of maximal emission and maximal absorption of their assembly with pFAST. The fluorescence quantum yield (ϕ), molar absorptivity at maximal absorbance (ε), and thermodynamic dissociation constant ($K_D$) values of their assembly with pFAST are indicated into brackets below the molecules. **b** Normalized absorption (left) and fluorescence (right) spectra of the different fluorogenic chromophores bound to pFAST. Spectra were recorded in pH 7.4 PBS at 25 °C. Fluorescence spectra were measured by exciting at the maximal absorption wavelength. **c** Solutions containing the different pFAST:chromophore assemblies illuminated with 365 nm light. Source data for spectra are provided as a Source Data file.

The variants oFAST, tFAST, and pFAST showed superior properties not only with the chromophore used for their selection, but with all chromophores of the HBO, HBT, and HBP series, demonstrating that our strategy successfully generated variants with increased chromophore-binding promiscuity. Unlike FAST, the three selected variants bind most of the chromophores of the HBO, HBT, and HBP series with submicromolar affinity (Fig. 2h, Supplementary Tables 2–4). This gain in affinity was accompanied with an increase of the fluorescence quantum yield of almost all fluorescent assemblies (Fig. 2h, Supplementary Tables 2–4). The three variants displayed also an improved affinity (up to one order

of magnitude) for the chromophores of the HBR series, in agreement with an increase of the chromophore binding promiscuity (Fig. 2h, Supplementary Tables 2–4).

Of the three selected variants showing superior performance, pFAST emerged as the best, and was thus tested with other chromophores. pFAST was shown to form tighter assemblies with HBR analogs bearing an additional carboxymethyl group on the rhodanine head (HBRAA series, Fig. 1 and Supplementary Table 4). These chromophores were previously shown to have a reduced cellular uptake because of their negatively charged carboxylate at physiological pH, allowing the specific labeling of

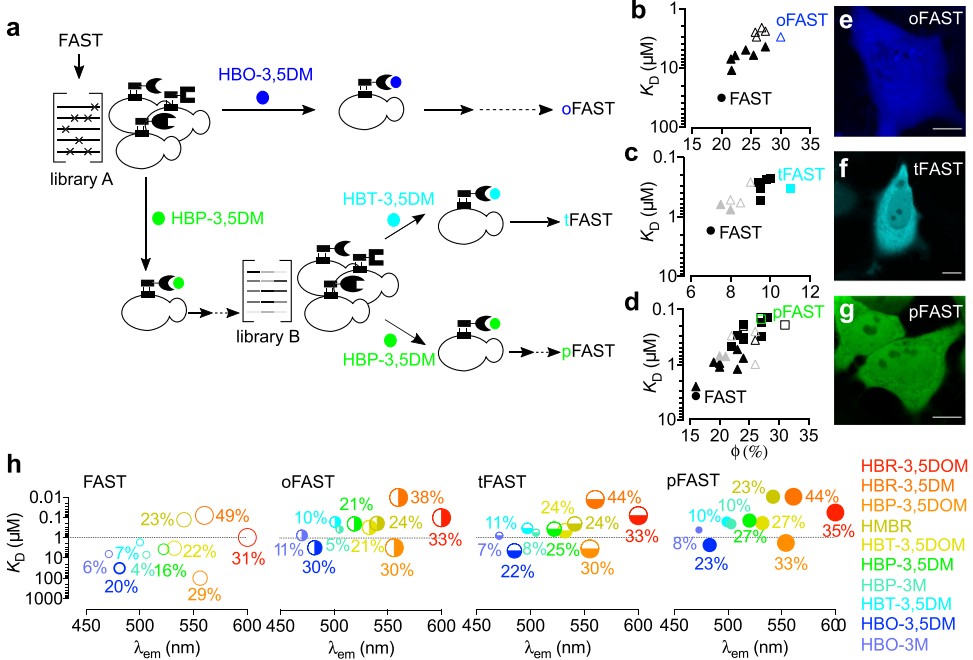

**Fig. 2 Engineering of promiscuous FAST variants. a** Directed evolution allowed the generation of oFAST, tFAST, and pFAST. Yeast-displayed libraries of variants of FAST were screened in presence of the indicated fluorogenic chromophore by fluorescence-activating cell sorting (solid arrows). Additional mutations were introduced by rational design (dotted arrows). **b–d** $K_D$s (thermodynamic dissociation constants) and fluorescent quantum yields (ϕ) of the clones isolated from the selections with **b** HBO-3,5DM, **c** HBT-3,5DM, and **d** HBP-3,5DM. Filled triangles are variants from library A; black squares are variants from library B; gray triangles are the variants used for the generation of the DNA shuffling library B and unfilled triangles and squares are variants generated by rational design. Values are also given for FAST for comparison. **e–g** Confocal micrographs of HeLa cells expressing cytoplasmic **e** oFAST labeled with 10 µM HBO-3,5DM, **f** tFAST labeled with 5 µM HBT-3,5DM and **g** pFAST labeled with 5 µM HBP-3,5DM. Experiments were repeated 1 time (**e**, **f**) and three times (**g**) with similar results. Scale bars 10 µm. See Supplementary Table 12 for imaging settings. **h** $K_D$s (thermodynamic dissociation constants) and emission wavelength of the different fluorogens in presence of FAST variants. Fluorescent quantum yields are given in %. The circle diameter reflects the value of the fluorescence quantum yield, while the different coloring systems are used to tell apart the different variants. Source data for graphs are provided as a Source Data file.

FAST-tagged proteins located at the cell surface[20]. In particular, we discovered that pFAST could efficiently interact with the unreported HBRAA-3,5DM, forming a bright orange fluorescent assembly that was 2.5-fold brighter than FAST:HBRAA-3E, the most efficient combination for selective labeling of cell surface proteins reported so far[20]. Finally, we discovered that pFAST could also bind substituted 4-hydroxybenzyldidene isorhodanine derivatives (HBIR series, Fig. 1) with single- or double-digit nanomolar affinities (Supplementary Table 4). The resulting assemblies exhibited red-shifted absorption and emission compared to their HBR counterparts. However, only the red fluorescent pFAST:HBIR-3,5DOM assembly exhibited a decent fluorescent quantum yield: the assemblies with HBIR-3M and HBIR-3,5DM, although among the tightest observed in this study, almost did not fluoresce. Although these properties would disqualify HBIR-3M and HBIR-3,5DM for fluorescence imaging applications, they allowed the efficient generation of absorbing-only dark assemblies (vide infra).

Overall, this set of experiments demonstrated that the selected pFAST was a highly promiscuous protein tag able to bind with high affinity a large variety of structurally related chromophores with diverse electronic properties. Our expanded collection of chromophores allowed the efficient generation of a palette of non-fluorescent and fluorescent assemblies emitting blue, cyan, green, yellow, orange, and red light using a single protein tag (Fig. 1a–c, Supplementary Fig. 5).

**Structural model.** Primary sequence alignments showed that the mutations K17N, G21E, G25R, A30V, Q41L, S72T, V83A, M95T,

M109L, S117R found in pFAST were mutation hotspots found in other variants (Supplementary Fig. 3). To get insights into the functional role of these mutations, atomic-resolution models of FAST and pFAST were generated by homology modeling using the crystal structure of the evolutionary related photoactive yellow protein (PYP) from *Halorhodospira halophila* as template (Fig. 3a and Supplementary Fig. 6a). The analysis of the root mean square fluctuation (RMSF) of each residue during molecular dynamic simulations showed an overall rigidification of the structure of pFAST compared to FAST (Fig. 3a, b). In particular, the loop 94-101 originally mutated to generate FAST from PYP, shows very little movement within pFAST compared to FAST. This overall rigidification was accompanied by a significant reduction of the accessible binding site volume compared to FAST and PYP (Supplementary Fig. 6a, b). All mutations found in pFAST participate to the rigidification of the overall structure and/or in the stabilization of the protein-chromophore assembly. The mutations G21E, G25R introduce two polar residues at the protein surface, which improves the solvation of the protein and thus the compacity of the structure (Supplementary Fig. 6e, f). The mutations A30V and Q41L enable strong hydrophobic interactions with residues in the N-terminal domain, known to be flexible in PYP, and thus play a major role in compacting and rigidifying the entire three-dimensional structure (Supplementary Fig. 6g, h). The mutation S72T creates interactions with the residues Glu 74, Phe75, Tyr76, which have a direct impact on the binding site structure (Supplementary Fig. 6i). The mutation V83A enables closer contacts with the residues Phe79, Ser85, Gly86, Arg87, and Tyr118, strengthening core interactions

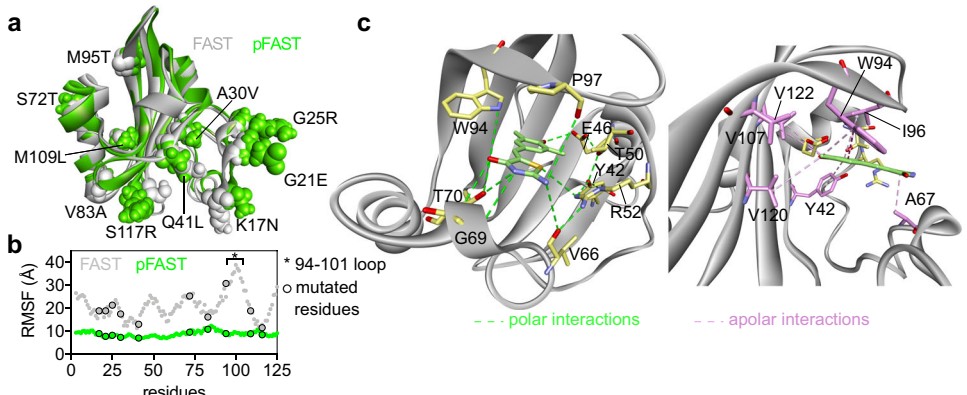

**Fig. 3 Structural model of pFAST. a** Structural model of pFAST and FAST generated by homology modeling using the crystal structure of the *Halorhodospira halophila* Photoactive Yellow Protein PYP (PDB: 6P4I) (See also Supplementary Fig. 6a). Mutations of pFAST relative to FAST are indicated. **b** Root Mean Square Fluctuation (RMSF) of the residues within pFAST and FAST during molecular dynamic simulations. **c** Polar and apolar interactions network involved in HBP-3,5DM binding and recognition within pFAST. Source data for graphs are provided as a Source Data file.

(Supplementary Fig. 6j). The mutation M95T strengthens the C-terminal β-sheet through additional interactions with Lys104 and Pro102 (Supplementary Fig. 6k). The mutation M109L directly impacts the size of the binding site by creating hydrophobic interactions with Phe75 and Val120 (Supplementary Fig. 6l). Finally, the mutation S117R participates to the overall rigidification of the structure by creating polar interactions with Asp34 in the N-terminal domain and apolar interactions with Ala112, strengthening the β-sheet structure (Supplementary Fig. 6m). The overall rigidification of pFAST structure creates a chromophore binding site that is smaller, but more open and better defined (Supplementary Fig. 6). Chromophore docking showed that pFAST was able to bind all chromophores of the HBO, HBT, HBP, and HBR series with higher affinities than FAST does, as observed experimentally (Supplementary Fig. 6c). Our model confirmed that chromophores adopt (i) a quasi-planar conformation and (ii) an orientation of the phenolic ring enabling the hydrogen-bond network formed by the conserved residues Tyr42, Glu46, Thr50, and Arg52 to stabilize the phenolate anion in the same way that PYP does with its chromophore[24–27], as originally proposed[13] (Fig. 3c, Supplementary Fig. 6n). This binding mode explain why chromophore binding induces both an increase of the fluorescence quantum yield (as the planar conformation favors radiative relaxation), and a strong red-shift in absorption (as the protonation/deprotonation equilibrium is shifted towards the deprotonated state). Our analysis showed that Glu46 forms a strong hydrogen bond with the phenolate, while Arg52 interacts with the electron-withdrawing heterocycle through hydrogen bonding with the endocyclic heteroatom (Fig. 3c and Supplementary Fig. 6n). In addition, Trp94 and Pro97, known to be essential for chromophore recognition and activation[13], form hydrogen bond and apolar interactions with the chromophore. Fine analysis with HBP-3,5DM allowed us to map the entire network of polar and apolar interactions involved in chromophore recognition and activation (Fig. 3c).

**Multicolor fluorescent labeling in mammalian cells**. Preliminary experiments showed that incubation with the fluorogenic chromophores for 24 h had no deleterious effects on mammalian cells at the concentrations used for labeling (Supplementary Fig. 7) and that the expression of the selected variants in mammalian cells was homogenous, in agreement with high intracellular stability (Fig. 2e–g). Imaging of live HeLa cells expressing pFAST fused to the histone H2B and treated for few tens of seconds with the different membrane permeant

fluorogenic chromophores demonstrated that pFAST could be used to efficiently encode blue, cyan, green, yellow, orange, and red fluorescence in mammalian cells (Fig. 4a). All chromophores showed good–excellent performances for high-contrast imaging, except HBO-3,5DM, and HBO-3M that formed dimmed blue fluorescent assemblies in agreement with their in vitro properties. Fluorescence quantification after labeling in both live and fixed cells enabled to show that fixation with paraformaldehyde did not significantly alter fluorescence, suggesting that pFAST remained functional after fixation and could still be labeled (Supplementary Fig. 8). We observed in particular that signal to background ratio in live and fixed cells did not significantly change no matter the chromophore used (Supplementary Fig. 8).

We next showed that selective labeling of cell-surface pFAST-tagged proteins could be efficiently achieved with the membrane-impermeant HBRAA-3M, HBRAA-3E, and HBRAA-3,5DM. In agreement with the low cell uptake of these three chromophores, labeling of a secreted transmembrane fusion with a pFAST domain facing the extracellular space led to a fluorescence signal restricted at the plasma membrane (Fig. 4a, Supplementary Fig. 9). We observed that pFAST outperformed FAST in terms of effective brightness in cells, and that HBRAA-3,5DM was the best candidate for the efficient labeling of pFAST-tagged cell-surface proteins, in agreement with its superior in vitro brightness.

**Photostability**. Continuous chromophore exchange can reduce the apparent rate of photobleaching through renewal of photodamaged chromophores. Photooxidation of specific methionine and tryptophan residues was however previously shown to cause FAST photodamage[28]. Long-term irradiation of live HeLa cells expressing pFAST showed that pFAST displayed enhanced photostability with HMBR, HBR-3,5DM, and HBR-3,5DOM, reaching photostability as high as the highly photostable enhanced GFP (EGFP) (Supplementary Fig. 10a–c). The mutations M95T and M109L may explain the higher photoresistance of pFAST compared to FAST. Prolonged imaging in presence of HBP-3,5DM, HBP-3,5DOM, HBP-3M, and HBT-3,5DOM showed that pFAST exhibited also good photostability with these fluorogenic chromophores, although a rapid transient fluorescence decay was observed before reaching steady state with the last three (Supplementary Fig. 10). Initial fluorescence could be fully recovered in the dark, demonstrating that this initial photobleaching was fully reversible. Decreasing light intensities or reducing imaging frequency enabled furthermore to reduce the amplitude of the initial fluorescence drop. This set of experiments suggested that

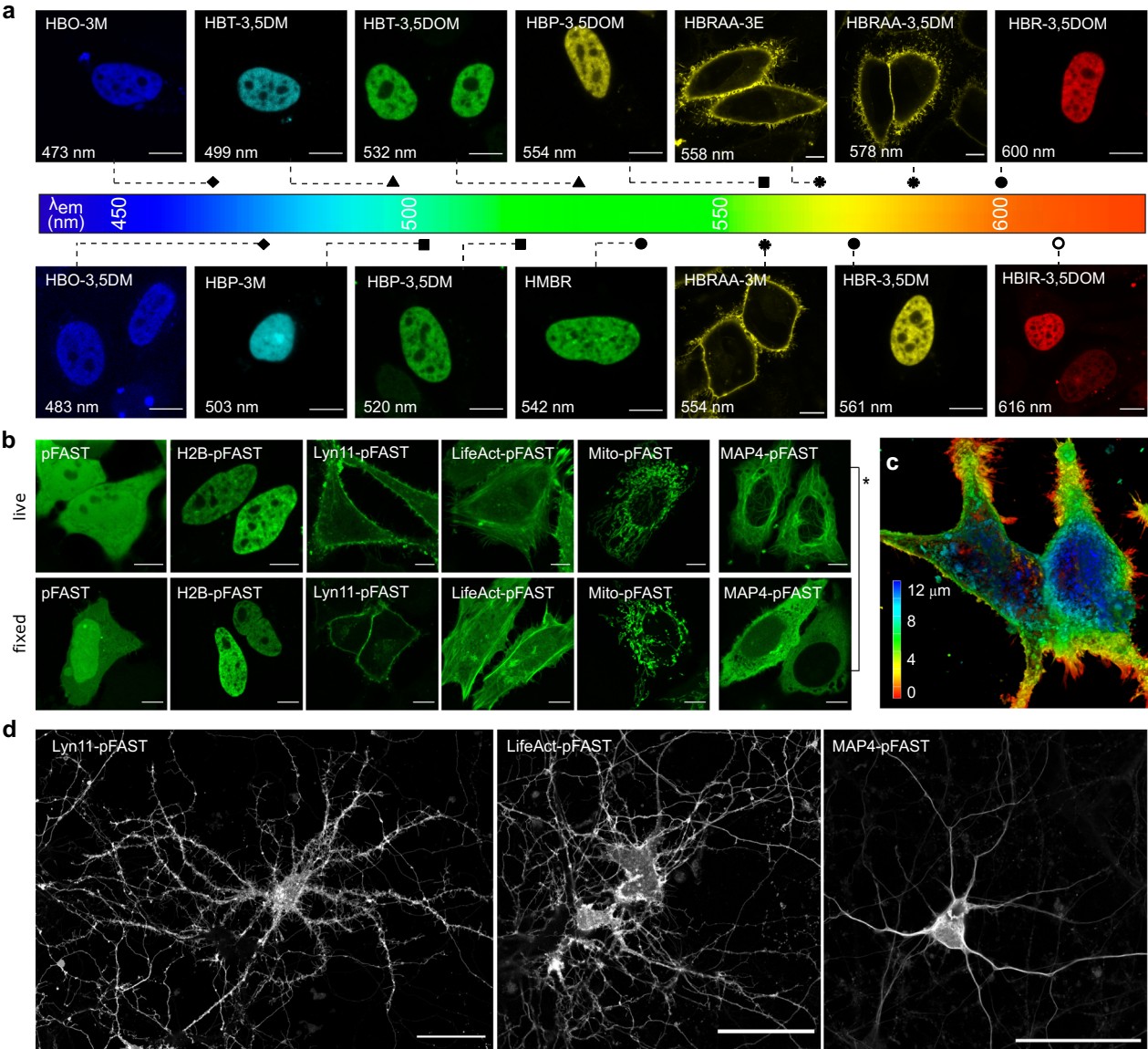

**Fig. 4 Selective imaging of pFAST fusions in various cellular localizations in mammalian cells and cultured neurons. a** Confocal micrographs of HeLa cells expressing pFAST fusions (either with H2B or to a transmembrane domain) in presence of the entire set of fluorogenic chromophores. Scale bars 10 μm. Experiments were repeated > 3 times with similar results. **b** Confocal micrographs of life and fixed HeLa cells expressing pFAST fused to: histone H2B, lyn11 (inner membrane-targeting motif), LifeAct (actin-binding peptide domain), mito (mitochondrial targeting motif), and to microtubule-associated protein (MAP) 4 and labeled with 5 μM HBP-3,5DM. Note that control experiments showed that the mislocalization of MAP4-pFAST observed in fixed cells (*) was due to MAP4 fixation rather than pFAST (see also Supplementary Fig. 12). Experiments were repeated >3 times with similar results. Scale bars, 10 μm. **c** Depth color coded three-dimensional reconstruction (from 81 optical sections over 12.8 μm) of live HeLa cells expressing lyn11-pFAST fusion and labeled with 5 μM HBR-3,5DOM. **d** Confocal micrographs of live dissociated hippocampal neurons expressing pFAST fused to lyn11, LifeAct, and MAP4 and labeled with 10 μM HBR-3,5DOM. Experiments were repeated >6 times with similar results. Scale bars, 50 μm. The Lyn11-pFAST image results from the maximum intensity projection of five optical sections. **a–d** See Supplementary Table 13 for imaging settings.

the observed reversible photobleaching was likely due to photo-isomerization and/or photoejection of the chromophore, as previously proposed to explain the reversible photobleaching observed when labeling FAST and its variants with low concentrations of HBR derivatives[23,28]. This singular photochemical signature might be used advantageously for selective imaging techniques based on kinetic discrimination[29,30] or for single-molecule localization microscopies[31,32].

**Imaging fusion proteins in mammalian cells and cultured neurons.** To verify that pFAST was well suited for the selective

imaging of proteins in various cellular localizations, HeLa and HEK293T cells expressing pFAST fused to the histone H2B (H2B-pFAST), the mitochondrial targeting sequence from the subunit VIII of human cytochrome C oxidase (Mito-pFAST), the inner plasma membrane targeting sequence lyn11 (Lyn11-pFAST), the actin binding peptide LifeAct (LifeAct-pFAST) and the microtubule-associated protein (MAP) 4 (MAP4-pFAST) were imaged by confocal microscopy in the presence of HBP-3,5DM. pFAST fusions showed proper cellular localization in live cells, demonstrating that pFAST did not perturb protein functions (Fig. 4b, and Supplementary Fig. 11). Similar results were observed in fixed cells, with the exception of MAP4-pFAST that

showed homogeneous fluorescence instead of microtubule-associated signal. (Fig. 4b). Control experiments using MAP4 fused to the yellow fluorescent protein mVenus or HaloTag gave similar mislocalization (Supplementary Fig. 12), suggesting that the observed mislocalization was an artifact induced by fixation. The high performance of pFAST further allowed us to generate high resolution three-dimensional reconstructions of live HeLa cells expressing lyn11-pFAST from 81 optical sections (Fig. 4c) showing the filopodia decorating plasma membrane contours.

To verify the suitability of pFAST for imaging proteins in delicate and sensitive cells, we next expressed FAST fusions in dissociated hippocampal neurons. Dissociated hippocampal neurons were successfully transfected with plasmids encoding Lyn11-pFAST, LifeAct-pFAST, or MAP4-pFAST. Live-cell confocal imaging showed (i) that labeled neurons remained fully viable with an impressive transfection rate (Supplementary Fig. 13), in agreement with neither pFAST fusions nor the fluorogenic chromophore being toxic, and (ii) that fusion proteins were properly localized (Fig. 4d). As observed for mammalian cells, pFAST allowed us to generate high-resolution confocal micrographs of live neurons (Fig. 4d) showing tiny but resolved protrusions, further demonstrating the high brightness and performance of pFAST for live imaging.

**Enhanced fluorescent labeling in multicellular organisms**. Mechanistic studies of most biological processes require imaging in multicellular organisms. Imaging of semisynthetic chemogenetic reporters in multicellular organisms is however often hampered by the low cell permeability of synthetic probes. HBR derivatives have been previously shown to allow the labeling of FAST and its derivatives in zebrafish embryos thanks to their very high cell uptake and superior ability to diffuse across multicellular systems[13,22,23]. Preliminary experiments in which we quantified the labeling efficiency in live HeLa cells by flow cytometry showed that full labeling of pFAST was achieved at lower chromophore concentrations than FAST, in agreement with its superior binding affinity (Supplementary Fig. 14), thus facilitating intracellular labeling and minimizing the amount of chromophore required for efficient detection. To verify that pFAST outperformed FAST in a similar fashion for imaging proteins in multicellular organisms, we compared the relative labeling efficiency of pFAST and FAST with various fluorogenic chromophores in chicken embryo. Comparison was performed in the same embryo by expressing each protein in a different side of the neural tube. Time-lapse imaging allowed us to monitor simultaneously the labeling of pFAST and FAST upon successive addition of chromophore solutions with gradually increased concentrations (Supplementary Fig. 15, Supplementary Movie 1). With all tested chromophores, pFAST reached full labeling at lower concentrations than FAST. Fluorescence saturation was furthermore reached within few tens of minutes, in agreement with the high cell uptake of the tested chromophores. The effective brightness of pFAST was in general comparable or higher than that of FAST, further demonstrating the superiority of pFAST over FAST for labeling proteins into tissue (Supplementary Fig. 15).

Encouraged by its superior performance in multicellular organisms, we next compared pFAST with the cyan fluorescent protein mCerulean, the green fluorescent protein EGFP, the yellow fluorescent protein EYFP and the orange fluorescent protein mKO. Direct comparisons were performed within the same embryo by expressing selectively each reporter in a different side of the neural tube. Apart from pFAST:HBP-3M that displayed lower effective brightness than mCerulean, as expected from their relative molecular brightness, all the other pFAST:-chromophore combinations showed performances comparable

with those of their fluorescent protein spectral equivalent (Fig. 5a–g, Supplementary Fig. 16).

Next, we demonstrated that pFAST was perfectly well suited for imaging proteins involved in dynamic cellular processes within the neuroepithelium of proliferating chicken neural tube by en-face time-lapse multicolor imaging. Tagging mitochondria with Mito-pFAST (labeled with HBR-3,5DOM) and membranes with iRFP670 (memb-iRFP670) allowed us to successfully visualize symmetrical mitochondrial segregation during cell division (Fig. 5h, Supplementary Movie S2). Expression of (i) H2B-pFAST (labeled with HBP-3,5DOM), and the PACT domain of pericentrin fused to mKO (pact-mKO) and (ii) H2B-pFAST (labeled with HMBR), pact-mKO, and memb-iRFP670 enabled us on the other hand to visualize chromosome segregation and centrosome dynamics during cell division (Fig. 5i, j, Supplementary Movies 3 and 4).

Finally, we exploited the large Stokes shift of pFAST:fluorogen assemblies to perform multicolor imaging using a single excitation and multiple detection windows. Large-Stokes-shift fluorescent probes can be combined with fluorescent probes displaying narrower Stokes shift to do multicolor imaging using a single excitation. The use of a single excitation allows a significant reduction of light exposure, and thus phototoxicity, which is highly beneficial for imaging in live tissues. We showed that the orange-red fluorescence of pFAST:HBR-3,5DOM could be efficiently detected upon excitation with 488 nm light, without any bleed-through of fluorescence emission in the channel for regular green fluorescent proteins, unlike the large-Stokes-shift fluorescent protein CyOFP1 (Supplementary Fig. 17a). This property allowed us to efficiently image the dynamics of EGFP fusions together with pFAST fusions in chicken embryo using a single 488 nm excitation (Supplementary Fig. 17b).

Beyond demonstrating that pFAST was well suited to image dynamic processes in living organisms, this set of experiments illustrates how the spectral properties of pFAST can be fine-tuned at will by using the most appropriate chromophore for a given experiment. Overall, these experiments demonstrated that pFAST allowed mechanistic studies of biological processes in multicellular organisms.

**Reversible labeling**. Some experiments require one to rapidly switch off fluorescence of a given fluorescent reporter. With fluorescent proteins, this can be done by photobleaching the reporter using intense light intensity, which can be toxic for cells and is challenging to perform in multicellular systems. As FAST labeling is non-covalent, it has been previously shown that washing away the fluorogenic chromophore by perfusing chromophore-free medium can reverse the labeling of FAST-tagged proteins in cells and switch off fluorescence in a simple way[13]. As HBIR-3M forms a very tight, non-fluorescent assembly with pFAST, we wondered if it could be used as a dark competitor capable of turning off pFAST fluorescence through chromophore exchange. Addition of an excess of HBIR-3M proved to efficiently switch off pFAST fluorescence in mammalian cells within just few seconds without the need to wash away the fluorogenic chromophore beforehand (Supplementary Fig. 18a–i, Supplementary Movie 5). Because of its single digit nanomolar binding affinity, HBIR-3M was able to efficiently replace fluorogenic chromophores binding pFAST with low, intermediate, and even high affinities. Interestingly, the fluorescence could be switch on again by replacement of the medium and addition of an excess of fluorogenic chromophore, enabling to control fluorescence at will. (Supplementary Fig. 18j–l, Supplementary Movie 6).

Encouraged by these results in cells, we next tested if HBIR-3M could allow one to switch off pFAST fluorescence in multicellular

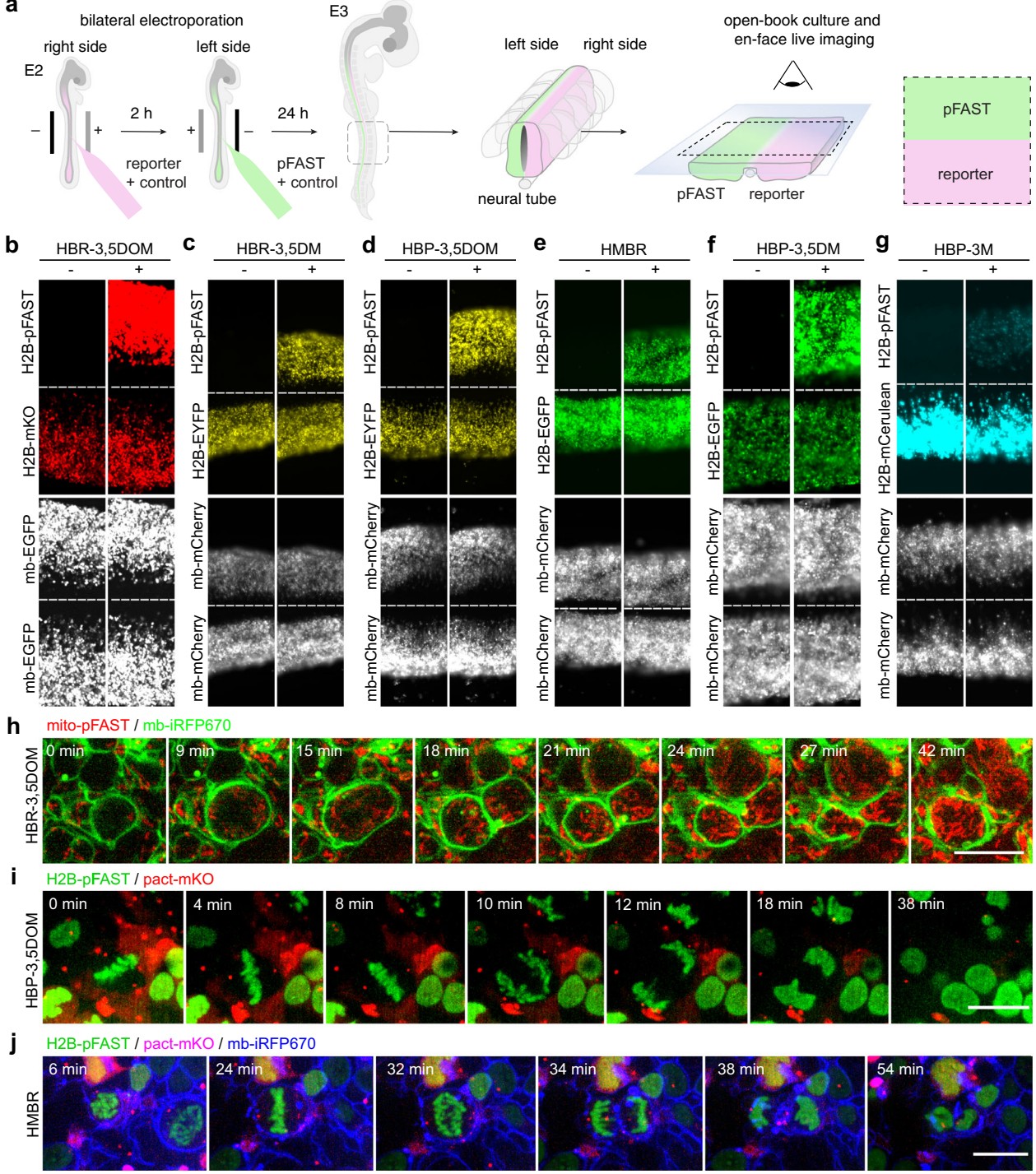

**Fig. 5 Selective imaging of pFAST fusions in chicken embryo. a–g** Plasmids encoding H2B-pFAST and **b** mKO, **c, d** EYFP, **e, f** EGFP, and **g** mCerulean fused to H2B were electroporated in each side of the neural tube in ovo at embryonic day 2 (E2, HH stage 13–14). mCherry or EGFP reporters were co-injected with each construct to monitor electroporation efficiency. 24 h later, embryos with homogeneous bilateral reporter expression in the neural tube were dissected and imaged before and after addition of the indicated fluorogenic chromophore. Experiments were repeated two times with similar results. **h–j** mito-pFAST (mitochondrial targeting motif) and memb-iRFP670 (membrane-targeting motif); **i** H2B-pFAST and pact-mKO (PACT domain of pericentrin) and **j** H2B-pFAST, pact-mKO, and mb-iRFP670 were electroporated in the neural tube in ovo, at embryonic day 2 (E2, HH stage 13–14). 24 h later, embryos were dissected, and imaged in presence of **h** 1 μM HBR-3,5DOM, **i** 5 μM HBP-3,5DOM, and **j** 1 μM HMBR using a spinning disk confocal microscope. Time-lapse showing cell division presented correct localization of the proteins (see also Supplementary Movies 2–4). Experiments in **h–j** were repeated 3, 2, and 2 times with similar results, respectively. Scale bars, 10 μm. **a–j** See Supplementary Table 13 for imaging settings.

systems, in which unlabeled through washing is more challenging because of the size of the specimen. Chicken neural tubes expressing H2B-pFAST were dissected and labeled with fluorogenic chromophores. After removal of the excess of chromophore, and washing with phosphate buffer, samples were treated with HBIR-3M and simply washed a second time. While simple washing was unsuccessful to switch off fluorescence, addition of an excess of HBIR-3M efficiently switched off pFAST within few tens of minutes (Supplementary Fig. 19, Supplementary Movie 7). Compared to protocols based on sequential washing steps, unlabeled through addition of a dark competitor can be a very simple and effective method to switch off fluorescence with high time resolution, in particular when slow passive diffusion prevents efficient washing in cells and multicellular organisms.

**FRET optimization and quantification**. FRET biosensors have become essential tools for studying signal transduction pathways in cells[33]. Their design and optimization remain challenging and labor-intensive because (i) various FRET pairs must be tested to obtain the best FRET signal, and (ii) proper characterization of FRET efficiency requires acceptor-free constructs to measure the loss of fluorescence intensity (or the change of lifetime) undergone by a donor when in close proximity of an acceptor. We anticipated that the FRET efficiency in biosensors containing pFAST as acceptor could be (i) easily optimized by testing various chromophores, and (ii) directly quantified in a single experiment by measuring the fluorescence loss or lifetime variation upon addition of the fluorogen. To demonstrate this, we fused pFAST to the cyan fluorescent protein mTurquoise2 or to the green fluorescent protein EGFP, and expressed the corresponding tandem constructs in HeLa cells. Quantification of the loss of fluorescence intensity by confocal microscopy upon addition of various chromophores showed that the fluorogenic HBR-3,5DOM chromophore and the dark absorbing-only HBIR-3M chromophore allowed to form the most efficient acceptors for mTurquoise2 and EGFP (Supplementary Fig. 20). Similar FRET measurements using fluorescence lifetime imaging microscopy (FLIM) confirmed the suitability of HBR-3,5DOM and HBIR-3M for FRET applications (Fig. 6a–e). Note that the ability to form a dark acceptor with HBIR-3M is advantageous for multiplexed experiments, as it allows one to use the imaging channel normally used by the acceptor for imaging another probe or sensor.

To further demonstrate the possibility of optimizing FRET biosensors, we adapted a previously developed aurora kinase A / AURKA biosensor. AURKA is a serine/threonine kinase that plays key roles during mitosis and undergoes a conformational change upon activation-induced autophosphorylation. FRET biosensors were previously generated by flanking AURKA with donor-acceptor FRET pairs. FRET increase allowed to demonstrate that AURKA was activated at centrosomes both at mitosis and during the G1 phase[34,35]. We flanked AURKA with mTurquoise2 and pFAST. As fluorogens can be applied sequentially, it was possible to measure in single U2OS cells synchronized at mitosis the FRET efficiency of the activated sensor at the mitotic spindle with HBR-3,5DOM and then with HBIR-3M by simple chromophore replacement (Fig. 6f–k). We could show that HBIR-3M was the best-suited chromophore in this context. The possibility to test various acceptors in the exact same cellular context through sequential labeling allows for optimizing FRET biosensors at the single-cell level. Overall, this set of experiments demonstrated that the tunability and versatility of pFAST enable to (i) quantify the efficiency of FRET biosensors in live cells, and (ii) optimize their properties through screening of the best acceptor by testing multiple fluorogens.

**Imaging fusion proteins below the diffraction limit**. Encouraged by the superior performance of pFAST, we further tested pFAST together with advanced confocal microscopy techniques. Airyscan confocal imaging is a powerful tool allowing the imaging of biological structures in live cells with enhanced signal-to-noise ratio and increased spatial resolution (up to 140 nm). Airyscan confocal imaging of live HeLa cells expressing Lyn11-pFAST or MAP4-pFAST labeled with various HBR or HBP derivatives allowed us to visualize details unresolved by traditional confocal microscopy (Supplementary Fig. 21). Airyscan imaging in live cells allowed us to image the rapid dynamics of microtubules and membranes with high temporal and spatial resolution (Supplementary Movie 8).

We extended our study to Stimulated Emission Depletion (STED) nanoscopy. STED microscopy is a powerful technique allowing the imaging of biological structures in fixed and live cells with nanoscale resolution. As many super-resolution microscopy techniques, the impact of STED nanoscopy is however limited by the number of compatible fluorophores. In conventional 2D-STED, a red-shifted doughnut-shaped STED pulse featuring zero intensity at the very center is applied immediately after the excitation pulse, resulting in depletion of the excitation state of the fluorophores within the doughnut area by stimulated emission. Maximal resolution is obtained with bright and photostable fluorophores that are highly sensitive to the STED laser for efficient depletion, and that are not re-excited by it[36]. Among our extended palette of chromophores, HBR-3,5DOM appeared well suited for STED as it allowed the generation of a bright and photostable red fluorescent reporter with an 80 nm Stokes shift (Excitation / Emission peaks at 520 nm / 600 nm) that allows effective depletion using the common 775 nm pulsed depletion laser while minimizing re-excitation. The use of a 3D depletion (3D-STED) was preferred over conventional 2D depletion (using doughnut-shaped depletion laser beam used in 2D-STED) in order to enhance both the lateral *and* axial resolution. Labeling with HBR-3,5DOM allowed the imaging of Lyn11-pFAST (Fig. 7a–d) and MAP4-pFAST (Fig. 7e–h and Supplementary Fig. 22) in live HeLa cells with improved resolution. pFAST:HBR-3,5DOM was bright and photostable enough to resist multiple average acquisition (16 average lines). 3D-STED microscopy allowed us to resolve fine cellular structures such as filopodia and microtubules in live cells (Fig. 7a–h and Supplementary Fig. 22). To compare the performance of pFAST:HBR-3,5DOM with other probes, we imaged in the same conditions pFAST:HBR-3,5DOM, the red fluorescent protein mCherry and HaloTag labeled with commercially available tetra-methylrhodamine (TMR). Whereas the three probes displayed comparable photostability in confocal microscopy (Supplementary Fig. 10d and Supplementary Fig. 23a–c), mCherry photobleached rapidly under 3D-STED conditions (Supplementary Fig. 23a) and did not allow us to generate images with higher resolution (Supplementary Fig. 23d). On the other hand, pFAST:HBR-3,5DOM and HaloTag-TMR showed higher resistance to photobleaching under 3D-STED conditions (Supplementary Fig. 23b, c), and enabled us to obtain STED images with comparable resolution improvement (Supplementary Fig. 23e, f). pFAST:HBR-3,5DOM having spectral properties blue-shifted compared to far-red fluorophores traditionally used with the 775 nm pulsed depletion laser, we next asked if we could perform two-color STED in live HeLa cells using a single 775 nm depletion laser. Labeling microtubules with MAP4-pFAST and filamentous actin with Silicon-Rhodamine(SiR)-actin allowed us to image these two cytoskeletal elements with improved resolution (Fig. 7i–k), demonstrating our ability to perform two-color STED imaging in live cells. Having at hand a chromophore with spectroscopic properties compatible with 3D-STED nanoscopy allowed us next to successfully image pFAST fusions below the diffraction barrier in live dissociated hippocampal neurons. Labeling actin with LifeAct-pFAST enabled us to resolve details of axons and

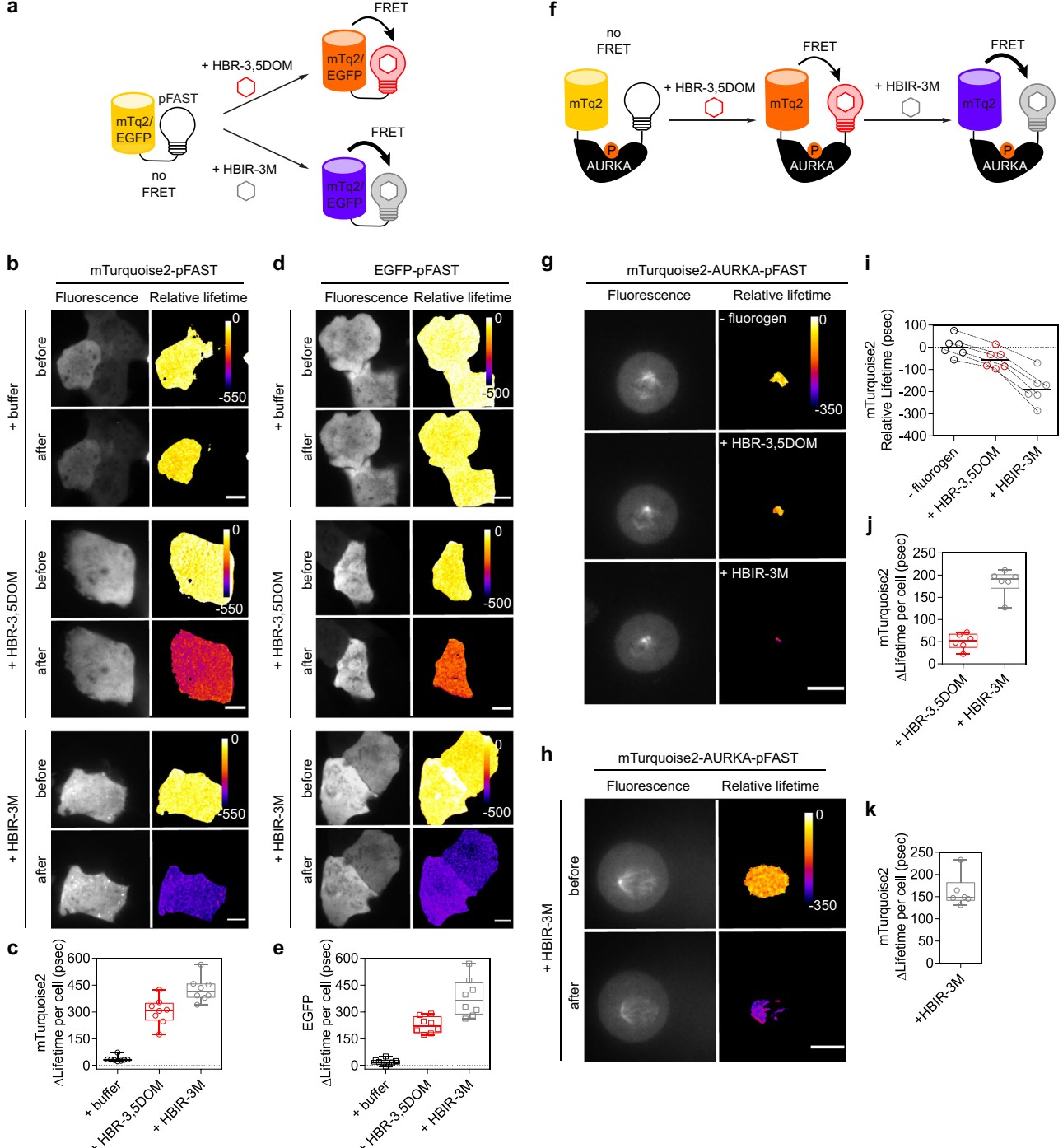

**Fig. 6 Fluorescence lifetime imaging of mTurquoise2 and EGFP donors using pFAST as acceptor in FRET experiments. a** Model illustrating the FRET experiments of mTurquoise2-pFAST and EGFP-pFAST tandems before and after addition of HBR-3,5DOM or HBIR-3M chromophores. **b–e** Representative fluorescence (left panels) and relative lifetime images (right panels) of live U2OS cells expressing **b, c** mTurquoise2-pFAST and **d, e** EGFP-pFAST before and after addition of imaging buffer, 10 μM of HBR-3,5DOM or 10 μM of dark chromophore HBIR-3M (mTurquoise2 and EGFP channels). Experiments were repeated eight times with similar results. The box plots in **c** and **e** show the variation of the lifetime of the donor (ΔLifetime) calculated per cell after addition of chromophores ($n = 8$ cells). Whiskers represent the highest and lowest values. **f** Model illustrating the mode of action of the autophosphorylated AURKA biosensor. The complete sequence of AURKA is located between the donor (mTurquoise2) and the acceptor (pFAST). When AURKA is autophosphorylated on Thr288, the kinase brings mTurquoise2 and pFAST in close proximity allowing FRET detection after addition of chromophores. Of note, the real three-dimensional orientations of the two reporters are not known. **g, h** Representative fluorescence (left panels) and relative lifetime images (right panels) of live U2OS cells synchronized in mitosis expressing mTurquoise2-AURKA-pFAST after **g** sequential addition of 1 μM of HBR-3,5DOM and 10 μM of dark-competitor HBIR-3M and **h** simple addition of 10 μM HBIR-3M (mTurquoise2 channel). Experiments were repeated six times with similar results. **i–k** The graphs show mTurquoise2 donor **i** relative lifetime per cell (median values are reported) and **j, k** ΔLifetime calculated per cell after addition of chromophores ($n = 6$ cells, box plots with whiskers representing the highest and lowest values). **b, d, g, h** All scale bars are 10 μm. The artificial 'fire' color represents pixel-by-pixel lifetimes. See Supplementary Table 13 for imaging settings. Source data for graphs are provided as a Source Data file.

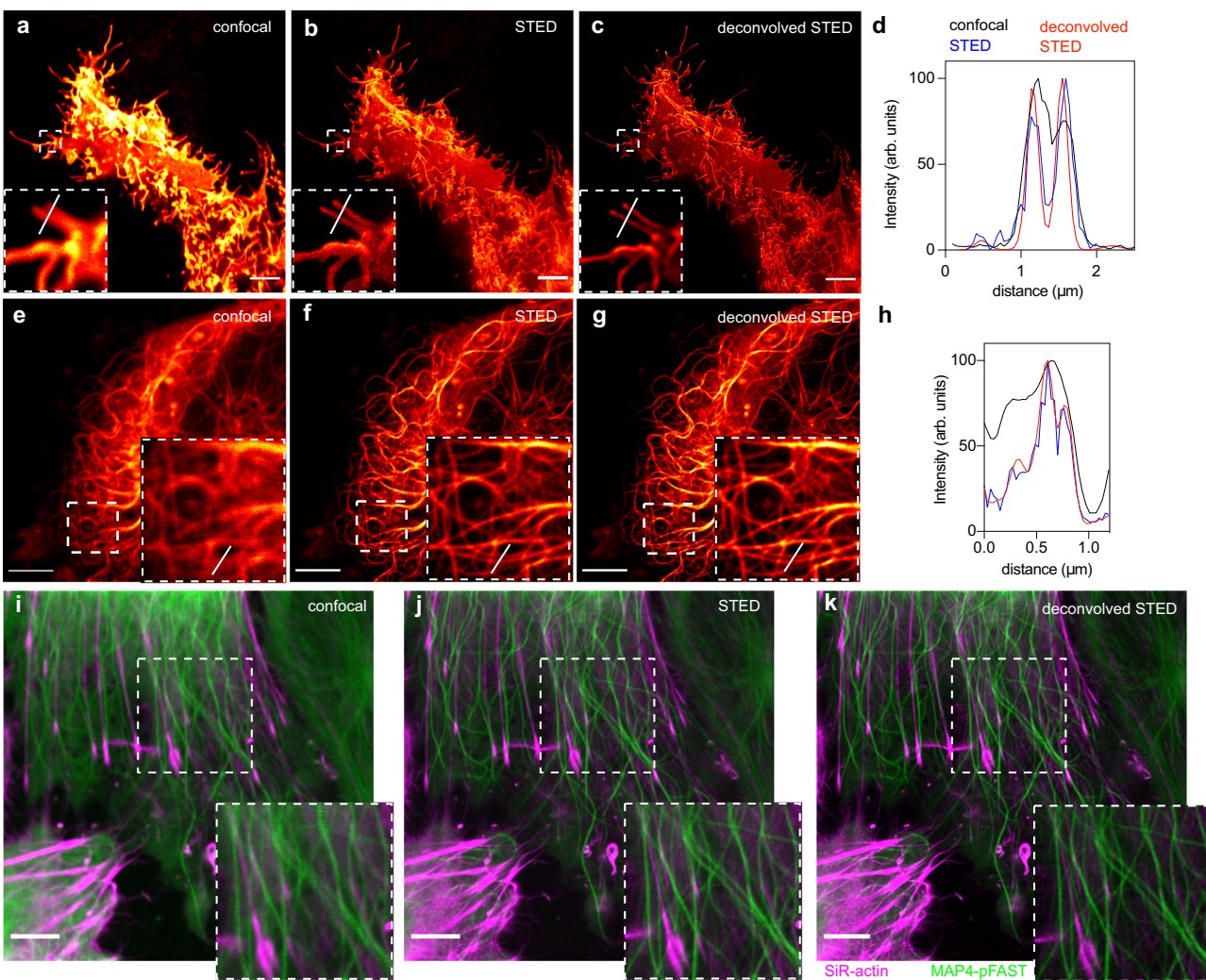

**Fig. 7 STED nanoscopy of pFAST-tagged proteins in live mammalian cells. a–h** Confocal and STED micrographs of live HeLa cells expressing **a–d** Lyn11-pFAST and **e–h** MAP4-pFAST. The graphs **d**, **h** show gain in resolution. Experiments with Lyn11-pFAST and MAP4-pFAST were repeated ten and 15 times, respectively, with similar results. **i–k** Two-color confocal and STED micrographs of live HeLa cells expressing MAP4-pFAST (green) and labeled with 1 μM Silicon-Rhodamine (SiR)-actin (magenta) using a single 775 nm depletion laser. Experiment was repeated ten times with similar results. Cells were treated with 10 μM HBR-3,5DOM before imaging. All scale bars are 5 μm (see Supplementary Table 13 for imaging settings). Source data for graphs are provided as a Source Data file.

dendrites, as well as visualize neurite growth cones with super-resolution (Fig. 8a–d). Similarly, targeting microtubule with MAP4-pFAST allowed us to image the microtubule network within live astrocytes with sub-diffraction resolution (Fig. 8e–g). Overall this set of experiments highlighted the potential of pFAST:HBR-3,5DOM for imaging proteins in live cells, including fragile neurons, with sub-diffraction resolution by STED nanoscopy.

## Discussion

Spectral tuning of chromophoric proteins can be achieved through modulation of the chemical structure of the embedded chromophore or by mutating amino acid residues in the chromophore vicinity. Here we report the successful engineering of pFAST, a fluorescent chemogenetic reporter with highly tunable spectral and optical properties. pFAST is a promiscuous variant of the protein tag FAST engineered to efficiently recognize a palette of chromophores exhibiting various electronic properties, and to form tight bimolecular assemblies with optical properties spanning the visible spectrum from blue to red. The use of isosteric electron-withdrawing heterocycles with various electronic properties allowed extensive spectral tuning with minimal structural changes. A massive engineering effort mostly conducted by directed evolution using a rapid and easy-to-implement fluorescence screening strategy based on yeast display and FACS further enabled us to develop a versatile tag showing good–excellent binding/spectroscopic properties with the entire set of fluorogenic compounds. In agreement with an increase of the chromophore binding promiscuity, pFAST showed improved properties with the chromophores of the prototypical FAST. In particular, efficient labeling in cells and in multicellular organisms could be achieved at lower chromophore concentration because of a gain in binding affinity, while longer or more power-demanding imaging experiments can be performed thanks to superior photostability. Structure modeling suggests that the ten mutations required to generate pFAST participate to the overall stabilization of the protein structure, leading to a smaller, more open, and better-defined chromophore binding pocket. Our modeling study further showed that the chromophore adopts a quasi-planar conformation and an orientation of the phenolic ring enabling the stabilization of the phenolate anion by a strong hydrogen bonding with Glu46 and the surrounding hydrogen bond network as found in PYP. This binding mode explains both

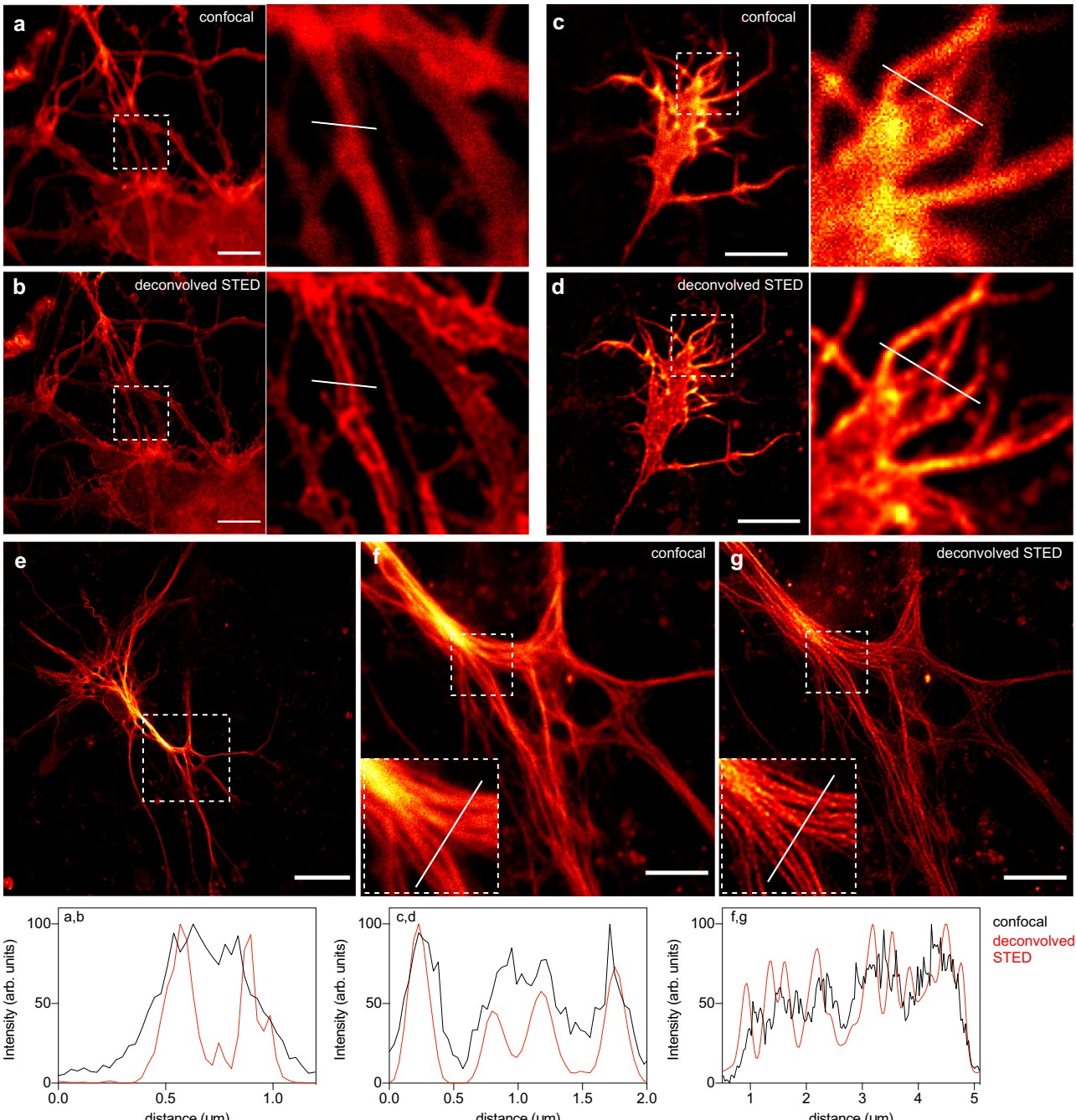

**Fig. 8 STED nanoscopy of pFAST-tagged proteins in live neurons and astrocytes. a, b** Confocal and STED micrographs of live dissociated hippocampal neuron expressing LifeAct-pFAST. **c, d** Confocal and STED micrographs of a neurite growth cone expressing LifeAct-pFAST. **e–g** Confocal and STED micrographs of a live astrocyte expressing MAP4-pFAST. Cells were treated with 10 µM of HBR-3,5DOM before imaging. Experiments with LifeAct-pFAST and MAP4-pFAST were repeated 10 and 20 times, respectively, with similar results. The graphs show gain in resolution. All scale bars are 5 µm, except the one in **e** that is 20 µm. **a–g** See Supplementary Table 13 for imaging settings. Source data for graphs are provided as a Source Data file.

the increase of fluorescence quantum yield upon binding, and the strong red-shift in absorption, as it was originally proposed[13] and recently shown[37] for FAST. Additional structural studies and reverse engineering would be necessary to fully decipher the role played by each mutation in the improvement of pFAST.

In this study, we showed that the extended chromophore promiscuity of pFAST provides high experimental versatility, allowing investigators to face a large variety of experimental scenarios. The panoply of spectrally distinguishable chromophores enables to fine tune pFAST spectral properties to optimize multicolor imaging. The large Stokes shift obtained with some fluorogens enables

moreover multicolor imaging using single excitation, reducing light exposure and thus phototoxicity. It should be noted that the spectral properties are however not yet fully optimal as the blue and green lights used for excitation may be toxic for cells and organisms. Future engineering efforts will focus on further red-shifting the absorption and emission properties in order to allow the use of less toxic red and far-red excitation lights for more biocompatible imaging as recently initiated with the engineering of far-red(fr)FAST that uses fluorogens with an elongated π system[22]. Selective labeling of cell-surface pFAST-tagged proteins can be furthermore efficiently achieved using bright membrane-impermeant fluorogenic chromophores. The

discovery of chromophores forming tight absorbing-only dark assemblies enables moreover to control the fluorescence state of pFAST. The possibility to replace bright chromophores with dark ones by direct competition for the binding site allows one to switch off pFAST fluorescence in cells with high time resolution.

The modularity and tunability of pFAST open also possibilities for FRET measurements. The ability to image with and without fluorogen provides advantages over fluorescent proteins, as it allows measurements with or without the acceptor dye in a single experiment. This unique feature provides an easy way to determine FRET efficiency at the single-cell level and eliminates the need for labor-intensive and time-consuming acceptor-free control experiments. Our collection of chromophores enables moreover to easily optimize a FRET system by testing various spectral properties for the acceptor using a single genetic construct. Application of multiple fluorogens in a sequential fashion can allow one to achieve biosensor optimization in single cells, allowing to compare biosensors in the exact same cellular context. The ability to form dark complexes absorbing in the green region opens furthermore interesting prospects for the design of dark acceptors for FRET measurements based on fluorescence lifetime imaging microscopy (FLIM). Like the dark acceptors ShadowG and ShadowY, previously engineered from GFP-like proteins[38,39], pFAST labeled with a dark chromophore does not fluoresce and can thus modulate the fluorescence lifetime of a donor without blocking an imaging channel.

Overall, our study demonstrates that pFAST can be a suitable molecular tool to image fusion proteins in live and fixed mammalian cells, including delicate cells such as primary cultured neurons, and in multicellular organisms such as chicken embryo. pFAST was successfully used to follow dynamic processes in real time in cells and in embryo tissues, demonstrating great potential for functional studies in cell biology. We demonstrated that pFAST was well suited to image cellular structures below the diffraction barrier in live mammalian cells and live cultured neurons using 3D-STED nanoscopy, whose broad use is currently limited by the low numbers of fluorophores displaying compatible spectral properties and photostability. The labeling of pFAST with HBR-3,5DOM allowed the successful imaging of membranes, microtubules, and actin filaments with sub-diffraction resolution. Beyond applications in STED nanoscopy, we anticipate that a complete and systematic study of the on-off fluorescence blinking or photoswitching behaviors of pFAST could open interesting prospects for other advanced imaging techniques, such as single-molecule tracking and super-resolution microscopy by single-molecule localization microscopy[32,40,41] or selective imaging exploiting the photokinetic dynamics of fluorescent probes[29,30].

## Methods
### Organic synthesis
*General.* Commercially available reagents were used as obtained. $^1$H and $^{13}$C NMR spectra were recorded at 300 K on a Bruker AM 300 spectrometer; chemical shifts are reported in ppm with protonated solvent as internal reference $^1$H, CHCl$_3$ in CDCl$_3$ 7.26 ppm, CHD$_2$SOCD$_3$ in CD$_3$SOCD$_3$ 2.49 ppm, CHD$_2$COCD$_3$ in CD$_3$COCD$_3$ 2.05 ppm; $^{13}$C, $^{13}$CDCl$_3$ in CDCl$_3$ 77.0 ppm, $^{13}$CD$_3$SOCD$_3$ in CD$_3$SOCD$_3$ 39.7 ppm, $^{13}$CD$_3$COCD$_3$ in CD$_3$COCD$_3$ 29.9 ppm; coupling constants $J$ are given in Hz. Mass spectra (chemical ionization and electronic impact with NH$_3$ or CH$_4$) were performed by the Service de Spectrométrie de Masse de Chimie ParisTech and mass spectra high resolution were performed by the Service de Spectrométrie de Masse de l'Institut de Chimie Organique et Analytique (Orléans). The preparation of HMBR (4-hydroxy-3-methylbenzylidene rhodanine), HBR-3,5DM (4-hydroxy-3,5-dimethylbenzylidene rhodanine), HBR-3,5DOM (4-hydroxy-3,5-dimethoxybenzylidene rhodanine), HBRAA-3M ((5-(4-hydroxy-3-methylbenzylidene)-4-oxo-2-thioxothiazolidin-3-yl)acetic acid) and HBRAA-3E (5-(4-hydroxy-3-ethylbenzylidene)-4-oxo-2-thioxothiazolidin-3-yl)acetic acid were previously described[13,20,21]. The starting compound oxazolidinedione was obtained according to previously described methods[42]. Synthetic schemes are presented on Supplementary Fig. 24.

**HBP-3M** ((4-hydroxy-3-methylbenzylidene)-2-iminothiazolidin-4-one). Solid 4-hydroxy-3-methylbenzaldehyde (130 mg, 1.0 mmol) and pseudothiohydantoin (69 mg, 0.6 mmol) were mixed and the mixture was brought to 170 °C. It first fused and then solidified. After cooling to 50–60 °C, ethanol was added and the mixture was heated at reflux. The solid partially disintegrated and formed a suspension. It was filtered and the remaining powder was dissolved in 1 M sodium hydroxide. The resulting solution was stirred for 5–10 min and was then slowly acidified with 1 M hydrochloric acid. After cooling to 0 °C, the precipitate was filtered, washed with cold water and dried over P$_2$O$_5$. HBP-3M was obtained as an orange powder (42%, 58 mg). $^1$H NMR (300 MHz, DMSO-d6, δ): 10.03 (s, 1H), 9.28 (s,1H), 9.00 (s,1H), 7.74 (s, 1H), 7.30 (d, 1H, $J$ = 2.0 Hz), 7.24 (dd, 1H, $J$ = 8.3 Hz, 2.0 Hz), 6.92 (d, 1H, $J$ = 8.3 Hz), 2.16 (s, 3H); $^{13}$C NMR (75 MHz, DMSO-d6, δ): 180.7, 175.4, 157.2, 132.2, 129.6, 128.9, 125.0, 124.8, 124.7, 115.3, 16.0; HRMS (ESI): m/z 235.0537 (calcd for C$_{11}$H$_{11}$N$_2$O$_2$S [M + H]$^-$ = 235.053).

**HBP-3,5DM** (4-hydroxy-3,5-dimethylbenzylidene)-2-iminothiazolidin-4-one). As HBP-3M using 4-hydroxy-3,5-dimethylbenzaldehyde (207 mg, 1.4 mmol) and pseudothiohydantoin (100 mg, 0.86 mmol). HBP-3,5DM was obtained as a dark orange powder (83%, 177 mg). $^1$H NMR (300 MHz, DMSO-d6, δ): 9.27(s,1H), 8.97(s,1H), 8.93 (s, 1H),7.42 (s, 1H), 7.16(s, 2H), 2.20 (s, 6H); $^{13}$C NMR (75 MHz, DMSO-d6, δ): 180.6, 175.4, 155.1, 130.1 (2 C), 129.7, 125.2, 124.9 (2 C), 16.7; HRMS (ESI): m/z 249.0694 (calcd mass for C$_{12}$H$_{13}$N$_2$O$_2$S [M + H]$^-$ = 249.0692).

**HBP-3,5DOM** ((4-hydroxy-3,5-dimethoxybenzylidene)-2-iminothiazolidin-4-one). As HBP-3M using 4-hydroxy-3,5-dimethoxybenzaldehyde (251 mg, 1.4 mmol) and pseudothiohydantoin (100 mg, 0.86 mmol). HBP-3,5DOM was obtained as an orange powder (61%, 146 mg). $^1$H NMR (300 MHz, DMSO-d6, δ): 9.31 (s,1H), 9.10 (s,1H), 9.03 (s,1H), 7.53 (s,1H), 6.88 (s,2H), 3.82 (s,6H); HRMS (ESI): m/z 281.0590 (calcd mass for C$_{12}$H$_{13}$N$_2$O$_4$S [M + H]$^-$ = 281.0591 (in accordance with previously described analysis[43])).

**HBT-3M** ((4-hydroxy-3-methylbenzylidene)thiazolidine-2,4-dione). Solid 4-hydroxy-3-methylbenzaldehyde (187 mg, 1.4 mmol) and 2,4-thiazolidinedione (100 mg, 0.85 mmol) were mixed and the mixture was brought to 170 °C. It first fused and then solidified. After cooling to 50–60 °C, ethanol was added to the solid and the suspension was stirred at reflux until complete dissolution. Then an excess of water was slowly added. The resulting precipitate was filtered, washed with water, and dried over P$_2$O$_5$. HBT-3M was obtained as a green powder (55%, 110 mg). $^1$H NMR (300 MHz, DMSO-d6, δ): 12.43 (s, 1H), 10.23 (s, 1H), 7.65 (s, 1H), 7.33(d, 1H, $J$ = 2.0 Hz), 7.29 (dd, $J$ = 8.3, 2.0 Hz, 1H), 6.92 (d, $J$ = 8.3 Hz, 1H), 2.16 (s, 3H); $^{13}$C NMR (75 MHz, DMSO-d6, δ): 168.2, 167.6, 158.2, 133.2, 132.5, 129.8, 125.2, 123.8, 118.7, 115.4, 15.9; HRMS (ESI): m/z 236.0376 (calcd mass for C$_{11}$H$_{10}$NO$_3$S [M + H]$^-$ = 236.0376).

**HBT-3,5DM** ((4-hydroxy-3,5-dimethylbenzylidene)thiazolidine-2,4-dione). As HBT-3M using 4-hydroxy-3,5-dimethylbenzaldehyde (205 mg, 1.4 mmol) and 2,4-thiazolidinedione (100 mg, 0.85 mmol). HBT-3,5DM was obtained as a yellow powder (75%, 154 mg). $^1$H NMR (300 MHz, DMSO-d6, δ): 12.43 (s, 1H), 9.14 (s, 1H), 7.61 (s, 1H), 7.18 (s, 2H), 2.20 (s, 6H); HRMS: m/z 250.0533 (calcd mass for C$_{12}$H$_{12}$NO$_3$S [M + H]$^-$ = 250.0532) (in accordance with previously described analysis[44,45]).

**HBT-3,5DOM** ((4-hydroxy-3,5-dimethoxybenzylidene)thiazolidine-2,4-one). As HBT-3M using 4-hydroxy-3,5-dimethoxybenzaldehyde (258 mg, 1.4 mmol) and 2,4-thiazolidinedione (100 mg, 0.85 mmol). HBT-3,5DOM was obtained as a green powder (61%, 146 mg). $^1$H NMR (300 MHz, DMSO-d6, δ): 12.48 (s, 1H), 9.33 (s, 1H), 7.71 (s, 1H), 6.89 (s, 2H), 3.81 (s, 6H); $^{13}$C (75 MHz, DMSO-d6, δ): 168.5, 167.9, 148.7, 139.1, 133.3, 123.7, 120.1, 108.4, 56.5; HRMS (ESI): m/z 282.0431 (calcd mass for C$_{12}$H$_{12}$NO$_5$S [M + H]$^-$: 282.0431) (in accordance with previously described analysis[44,46]).

**HBO-3M** ((4-hydroxy-3-methylbenzylidene)oxazolidine-2,4-dione). A mixture of 4-hydroxy-3-methylbenzaldehyde (100 mg, 0.74 mmol), 2,4-oxazolidinedione (148 mg, 1.5 mmol), and pyrrolidine (60 μL, 0.74 mmol) in ethanol (2 mL) was stirred at reflux for 18 h. After cooling, the mixture was diluted with water. The precipitate was collected by filtration, washed with cold water, and dried over P$_2$O$_5$. HBO-3M was obtained as an orange powder (48%, 78 mg). $^1$H NMR (300 MHz, DMSO-d6, δ): 12.16 (s, 1H), 10.04 (s, 1H), 7.51 (d, 1H, $J$ = 3 Hz), 7.47 (dd, 1H, $J$ = 3, 9 Hz), 6.86 (d, 1H, $J$ = 9 Hz), 6.58 (s, 1H), 2.14 (s, 3H); $^{13}$C NMR (75 MHz, DMSO-d6, δ): 165.9, 157.2, 154.3, 137.8, 133.1, 129.9, 124.6, 122.4, 115.1, 109.7, 15.9; HRMS (ESI): m/z 220.060512 (calcd mass for C$_{11}$H$_{10}$NO$_4$ [M + H]$^-$ = 220.060434).

**HBO-3,5DM** ((4-hydroxy-3,5-dimethylbenzylidene)oxazolidine-2,4-dione). As HBO-3M using 4-hydroxy-3,5-dimethylbenzaldehyde (100 mg, 0.66 mmol), 2,4-oxazolidinedione (134 mg, 1.3 mmol), pyrrolidine (55 μL, 0.66 mmol) in ethanol (2 mL). HBO-3,5DM was obtained as an orange powder (43%, 28%). $^1$H NMR (300 MHz, DMSO-d6, δ): 12.10 (s, 1H), 8.95 (s,1H), 7.37 (s, 2H), 6.53 (s, 1H), 2.19 (s, 6H); HRMS (ESI): m/z 234.076560 (calcd mass for C$_{12}$H$_{12}$NO$_4$ [M + H]$^-$ = 234.076084). (in accordance with previously described analysis[47])

**HBIR-3M** ((4-hydroxy-3-methylbenzylidene)-4-thioxothiazolidin-2-one). As HBP-3M using 4-hydroxy-3-methylbenzaldehyde (164 mg, 1.2 mmol) and isorhodanine (100 mg, 0.75 mmol). HBIR-3M was obtained as an orange powder (49%, 93 mg). $^1$H NMR (300 MHz, DMSO-d6, δ): 10.53 (s, 1H), 8.01 (s, 1H), 7.43 (d, 1H, $J$ = 2.0 Hz), 7.38 (dd, 1H, $J$ = 8.3 Hz, 2.0 Hz), 6.96 (d, 1H, $J$ = 8.3 Hz), 2.16 (s, 3H); $^{13}$C NMR (75 MHz, DMSO-d6, δ): 194.9, 170.8, 159.2, 137.3, 134.1, 130.8, 125.7, 125.6, 124.4, 115.7, 15.9; HRMS: m/z 252.014671 (calcd for C$_{11}$H$_{10}$NO$_2$S$_2$ [M + H]$^-$ = 252.014747).

**HBIR-3,5DM** ((4-hydroxy-3,5-dimethylbenzylidene)-4-thioxothiazolidin-2-one). As HBP-3M using 4-hydroxy-3,5-dimethylbenzaldehyde (180 mg, 1.2 mmol) and isorhodanine (100 mg, 0.75 mmol). HBIR-3,5DM was obtained as a brown powder (69%, 137 mg). $^1$H NMR (300 MHz, DMSO-d6, δ): 13.66 (s, 1H), 9.39 (s,

1H), 7.97 (s, 1H), 7.28 (s, 2H), 2.21 (s, 6H); $^{13}$C NMR (75 MHz, DMSO-d6, δ): 194.9, 170.8, 157.1, 137.2, 131.8, 125.7, 125.5, 124.5, 16.6; HRMS (ESI): m/z 266.030241 (calcd mass for $C_{12}H_{12}NO_2S_2$ [M + H]$^-$ = 266.030397).

**HBIR-3,5DOM** ((4-hydroxy-3,5-dimethoxylbenzylidene)-4-thioxothiazolidin-2-one). As HBP-3M using 4-hydroxy-3,5-dimethoxybenzaldehyde (220 mg, 1.2 mmol) and isorhodanine (100 mg, 0.75 mmol). HBIR-3,5DOM was obtained as a brown powder (48%, 110 mg). $^1$H NMR (300 MHz, DMSO-d6, δ): 13.72 (s, 1H), 9.60 (s, 1H), 8.05 (s, 1H), 6.97 (s, 2H), 3.83 (s, 6H); $^{13}$C NMR (75 MHz, DMSO-d6, δ): 194.9, 170.8, 148.3, 139.3, 137.5, 126.5, 123.8, 108.7, 56.1; HRMS (ESI): m/z 298.020265 (calcd mass for $C_{12}H_{12}NO_4S_2$ [M + H]$^-$ = 298.020226) (in accordance with previously described analysis)[48].

**HBRAA-3,5DM** 5-((4-hydroxy-3,5-dimethylbenzylidene)-4-oxo-2-thioxothiazolidin-3-yl) acetic acid. A solution of rhodanine-3-acetic acid (191 mg, 1.0 mmol) and 4-hydroxy-3,5-dimethylbenzaldehyde (150 mg, 1.0 mmol) in water (70 mL) was stirred at 90 °C for 7 days. After cooling to 4 °C and standing overnight, the precipitate was filtered through a glass filter and the crude solid was washed with water, ethanol and dried over $P_2O_5$, to give the desired product as an orange powder (160 mg, 50%). $^1$H NMR (300 MHz, CD$_3$SOCD$_3$, δ): 9.40 (s, 1H), 7.70 (s, 1H), 7.27 (s, 2H), 4.72 (s, 2H), 2.23 (s, 6H); $^{13}$C NMR (75 MHz, CD$_3$SOCD$_3$, δ): 193.6, 167.8, 166.9, 157.7, 135.3, 132.4(2C), 126.0(2C), 124.3, 117.4, 45.4, 17.1(2C); MS (ESI): m/z 322.2 (calcd mass for $C_{14}H_{12}NO_4S_2$ [M-H]$^-$ = 322.0).

**Biology.** The presented research complies with all relevant ethical regulations.

**General.** Synthetic oligonucleotides used for cloning were purchased from Integrated DNA technology. PCR reactions were performed with Q5 polymerase (New England Biolabs) in the buffer provided. PCR products were purified using QIA-quick PCR purification kit (Qiagen). The products of restriction enzyme digests were extracted and purified by preparative gel electrophoresis followed by QIA-quick gel extraction kit (Qiagen). Restriction endonucleases, DNAse I, T4 ligase, Fusion polymerase, Tag ligase and Tag exonuclease were purchased from New England Biolabs and used with accompanying buffers and according to manufacturer protocols. Isothermal assemblies (Gibson assembly) were performed using homemade mix prepared according to previously described protocols[49]. Small-scale isolation of plasmid DNA was done using QIAprep miniprep kit (Qiagen) from 2 mL overnight bacterial culture supplemented with appropriate antibiotics. Large-scale isolation of plasmid DNA was done using the QIAprep maxiprep kit (Qiagen) from 150 mL of overnight bacterial culture supplemented with appropriate antibiotics. All plasmid sequences were confirmed by Sanger sequencing with appropriate sequencing primers (GATC Biotech). All the plasmids used in this study are listed in Supplementary Tables 10 and 11. The protein and DNA sequences of FAST and pFAST are given in Supplementary Table 12.

**Yeast display**
*Library construction.* The first yeast library (called library A in this study) of FAST constructed by error-prone PCR using Genemorph II kit (Agilent) was previously described[23]. The second library (called library B in this study) was constructed by DNA shuffling according to different DNA shuffling library protocols[50–52]. Five gene variants (with more than 70% of sequences homology) previously screened, selected, and rationally designed from the library A were first amplified by PCR. They were randomly digested by DNAse I (20 U/mL). The digested products were purified and mixed together. Two steps of PCR amplification (one without primers and the second one with two primers to amplify the full-length genes) were then performed enabling random recombination and amplification of the full-length shuffled product. In addition, a mutation rate of 4 nt/gene and a deletion rate of 0.25 nt/gene were obtained (as shown by Sanger sequencing of 17 individual clones). The shuffled product was cloned in pCTCON2 using NheI and BamHI restriction sites. Large-scale transformation into DH10B was performed, yielding $5.5 \times 10^7$ transformants. The DNA was maxiprepped and transformed into yeast strain EBY100 (kind gift from K. D. Wittrup) using a large-scale, high-efficiency protocol[53] to yield $6.7 \times 10^7$ transformants.

*Selection.* The yeast library (about $1–5 \times 10^9$ cells) was grown overnight at 30 °C in 1 L SD medium (20 g/L dextrose, 6.7 g/L yeast nitrogen base, 1.92 g/L yeast synthetic dropout without tryptophan, 7.44 g/L NaH$_2$PO$_4$, 10.2 g/L Na$_2$HPO$_4$-7H$_2$O, 1% penicillin–streptomycin 10,000 U/mL). The following morning yeast culture was diluted to OD$_{600nm}$ 1 in 1 L of SD and grown at 30 °C until OD$_{600nm}$ 2–5. Next, $5 \times 10^6$ cells were pelleted and resuspended in 1 L of SG medium (20 g/L galactose, 2 g/L dextrose 6.7 g/L yeast nitrogen base, 1.92 g/L yeast synthetic dropout without tryptophan, 7.44 g/L NaH$_2$PO$_4$, 10.2 g/L Na$_2$HPO$_4$-7H$_2$O, 1% penicillin–streptomycin 10,000 U/mL) to grow for 36 h at 23 °C. $5 \times 10^8$ induced yeast cells were collected ($2500 \times g$—2 min), washed with 10 mL PBS (0.05 M phosphate buffer, 0.150 M NaCl, pH 7.4) + BSA (bovine serum albumin 1 g/L) and incubated for 30 min at room temperature with 1:250 dilution of primary antibody chicken anti-c-myc IgY (Life technologies) in 200 µL of PBS. Cells were then centrifuged, washed with PBS-BSA, and incubated for 20 min on ice with 1:100 dilution of secondary goat-anti chicken antibody coupled to AlexaFluor 647 in 200 µL of PBS. After centrifugation and washing with 10 mL PBS + BSA, the cells were resuspended in 5 mL of PBS supplemented with appropriate fluorogen

concentration. The cells were sorted on a MoFlo$^{TM}$ Astrios Cell sorter (Beckman Coulter) equipped with 405 nm, 488 nm, and 640 nm lasers. The fluorescence of HBO-3,5DM and HBO-3M binders were detected using the following parameters: Ex 405 nm, Em 458 ± 30 nm. For the selection using library A, the fluorescence of HBT-3,5DM and HBT-3M binders were detected using the following parameters: Ex 405 nm, Em 488 ± 5 nm. For the selection using library B, the fluorescence of HBT-3,5DM binders was detected using the following parameters: Ex 405 nm, Em 513 ± 13 nm. The fluorescence of HBP-3,5DM and HBP-3M binders were detected using the following parameters: Ex 488 nm, Em 526 ± 26 nm. Finally, the fluorescence of HBP-3,5DOM binders was detected using the following parameters: Ex 488 nm, Em 546 ± 10 nm. Sorted cells were collected in SD, grown overnight at 30 °C, and plates on SD plates (SD supplemented with 182 g/L D-sorbitol and 15 g/L agar) for ~60 h at 30 °C. The resulting lawn was collected in SD supplemented with 30% glycerol, aliquoted and frozen or directly used for the next round of selection. In total, 5–8 rounds of selection were performed for each fluorogen. After each selection, 24 clones per final rounds were isolated, screened by flow cytometry, and their plasmid DNA was isolated using miniprep kit (Qiagen), transformed in DH10B, and re-isolated for sequencing.

*Cloning.* Variants obtained by rational design were cloned in pCTCON2 plasmids driving EBY100 yeast surface expression using NheI and BamHI restriction sites as for library construction *(vide supra)*. All the plasmids used in this study are listed in Supplementary Table 10.

*Flow cytometry analysis.* Flow cytometry was performed on a Gallios analyzer (Beckman Coulter) equipped with 405 nm, 488 nm, and 638 nm lasers and ten filters and channels. To prepare samples for flow cytometry, small-scale cultures were grown as for library expression *(vide supra)*. Briefly, 5 mL of SD were inoculated with a single colony and grown overnight at 30 °C. The following day, the cultures were diluted in 5 mL of SD to a final OD$_{600nm}$ 1 and grown until OD$_{600nm}$ 2–5. These cultures were used to inoculate 5 mL of either SD (non-induced cultures) or SG (induced cultures) to an OD$_{600nm}$ of 0.5 and the cultures were grown for 36 h at 23 °C. The cultures were collected ($2500 \times g$—2 min) to reach a final concentration of $1 \times 10^8$ cells/mL. For analysis, the monoclonal yeast cells were prepared and incubated with the set of antibodies as for library expression (vide supra). The cultures were finally resuspended in PBS supplemented with appropriate fluorogens concentrations. Data were analyzed using Kaluza Analysis software (Beckman Coulter).

**Characterization of variants**
*Cloning.* Plasmids driving E. coli expression of the variants with an N-terminal His-tag under the control of a T7 promoter were constructed by replacing the sequence coding for FAST in the plasmid pAG87[13] using isothermal Gibson assembly. Site-directed mutagenesis was performed by isothermal Gibson assembly using primers with the mutations of interest. All the characterized variants are listed in Supplementary Table 11.

*Expression.* The plasmids were transformed in Rosetta (DE3) pLys E. coli (New England Biolabs). Cells were grown at 37 °C in lysogen Broth (LB) medium supplemented with 50 µg/ml kanamycin and 34 µg/mL chloramphenicol to OD$_{600nm}$ 0.6. Expression was induced overnight at 16 °C by adding isopropyl-β-D-1-thio-galactopyranoside (IPTG) to a final concentration of 1 mM. Cells were harvested by centrifugation ($4300 \times g$ for 20 min at 4 °C) and frozen.

*Purification.* The cell pellet was resuspended in lysis buffer (PBS supplemented with 2.5 mM MgCl$_2$, 1 mM of protease inhibitor PhenylMethaneSulfonyl Fluoride PMSF, 0.025 mg/ml of DNAse, pH 7.4) and sonicated (5 min, 20% of amplitude) on ice. The lysate was incubated for 2–3 h on ice to allow DNA digestion by DNAse. Cellular fragments were removed by centrifugation ($9000 \times g$ for 1 h at 4 °C). The supernatant was incubated overnight at 4 °C by gentle agitation with pre-washed Ni-NTA agarose beads in PBS buffer complemented with 20 mM of imidazole. Beads were washed with 10 volumes of PBS complemented with 20 mM of imidazole and with 5 volumes of PBS complemented with 40 mM of imidazole. His-tagged proteins were eluted with 5 volumes of PBS complemented with 0.5 M of imidazole. The buffer was exchanged to PBS (0.05 M phosphate buffer, 0.150 M NaCl) using PD-10 or PD-MidiTrap G-25 desalting columns (GE Healthcare). Purity of the proteins were evaluated using SDS-PAGE electrophoresis stained with Coomassie blue.

*Physico-chemical measurements.* Steady state UV-Vis absorption spectra were recorded using a Cary 300 UV-Vis spectrometer (Agilent Technologies), equipped with a Versa20 Peltier-based temperature-controlled cuvette chamber (Quantum Northwest), and fluorescence spectra were recorded using an LPS 220 spectrofluorometer (PTI, Monmouth Junction, NJ), equipped with a TLC50TM Legacy/PTI Peltier-based temperature-controlled cuvette chamber (Quantum Northwest). Fluorescence quantum yield measurements were determined as previously described using either FAST:HMBR or quinine sulfate as a reference[13,21,23,54]. Reciprocal dilution with protein solution was used so as to keep the protein concentration constant at 40 µM while diluting the fluorogen solution. Absorption coefficients were determined by forward titration of fluorogens into a 40 µM protein solution. Thermodynamic dissociation constants were

determined by titration experiments in which we measured the fluorescence of the fluorescent assembly at various fluorogen concentrations using a Spark 10 M plate reader (Tecan) and fitting data in Prism 9 to a one-site specific binding model. The thermodynamic dissociation constants ($K_D$) of the dark HBIR-3M and HBIR-3,5DM chromophores were determined by determining the apparent dissociation constant of HBP-3,5DM in presence of various concentrations of dark competitors (See Supplementary Note 2).

## Modeling

*Homology modeling.* The homology models of FAST and pFAST were generated according to models previously described[55,56]. Briefly, sequence alignments between FAST and pFAST and the ultra-high resolution structure of the *Halorhodospira halophila* Photoactive Yellow Protein (PYP) (Protein Data Bank (PDB) ID: 6P4I) were generated with Clustal W[57]. Alignments were manually refined to avoid gaps in predicted (FAST and pFAST) and known (PYP) secondary structure elements. Three-dimensional FAST and pFAST models were built from these alignments and from crystallographic atomic coordinates of PYP using the automated comparative modeling tool MODELER (Sali and Blundell) implemented in Discovery Studio. The best model according to DOPE score (Discrete Optimized Protein Energy) and potential energy calculated by modeler were solvated (10 Å water box and 0.145 M NaCl) and minimized using Adopted Basis NR algorithm to a final gradient of 0.001. The resulting structure were submitted to a 10 ns NAMD dynamic.

*Molecular docking.* Flexible ligand-rigid protein docking was performed using CDOCKER implemented in Discovery Studio 2019[58]. Random ligand conformations were generated from the initial ligand structure through high-temperature molecular dynamics. The best poses according to their ligscore2[59] were retained and clustered according to their binding mode. The most significant poses were solvated and minimized using Adopted Basis NR algorithm to a final gradient of 0.001.

## Experiments in mammalian cells

*General.* HeLa cells (ATCC CRM-CCL2) were cultured in Minimal Essential Media (MEM) supplemented with phenol red, Glutamax I, 1 mM of sodium pyruvate, 1% (vol/vol) of non-essential amino-acids and 10% (vol/vol) fetal calf serum (FCS), at 37 °C in a 5% $CO_2$ atmosphere. HEK 293 T (ATCC CRL-3216) cells were cultured in Dulbecco's Modified Eagle Medium (DMEM) supplemented with phenol red and 10% (vol/vol) fetal calf serum at 37 °C in a 5% $CO_2$ atmosphere. For imaging, cells were seeded in µDish IBIDI (Biovalley) coated with poly-L-lysine. Cells were transiently transfected using Genejuice (Merck) or Lipofectamine 2000 (Thermo Fisher Scientific) according to the manufacturer's protocols for 24–48 h prior to imaging. Live cells were washed with DPBS (Dulbecco's Phosphate-Buffered Saline), and treated with DMEM media (without serum and phenol red) containing the fluorogens at the indicated concentration. The cells were imaged directly without washing. Fixation of cells was performed using formaldehyde solution at 3.7% for 30 min. Fixed cells were then washed three times with DPBS and treated with DPBS containing the fluorogens at the indicated concentration prior to imaging. For FLIM-FRET experiments, U2OS cells (ATCC HTB-96) were cultured in Dulbecco's modified Eagle's medium (DMEM) supplemented with phenol red and 10% (vol/vol) fetal calf serum and 1% (vol/vol) penicillin–streptomycin at 37 °C in a 5% $CO_2$ atmosphere, plated on 8-wells Labtek (Nunc) and then imaged in phenol-red-free Leibovitz's L-15 medium (L-15, Thermo Fisher Scientific), supplemented with 20% fetal bovin serum, 1% L-glutamine, and 1% penicillin–streptomycin. For experiments using AURKA biosensor, mitotic cells were obtained after synchronization at the G2/M transition with 100 ng/mL nocodazole (Sigma-Aldrich) for 16 h. Cells were washed twice and incubated with prewarmed imaging medium containing the fluorogens at the indicated concentration for 30 min to reach metaphase.

*Cloning.* The plasmids allowing the mammalian expression of pFAST, tFAST and oFAST variants under the control of a CMV promoter were constructed by replacing the sequence coding for FAST[13] or iFAST[54] in the previously described plasmids pAG104 (FAST), pAG106 (lyn11-FAST), pAG109 (H2B-FAST), pAG156 (mito-FAST), pAG470 (lifeAct-iFAST) and pAG498 (MAP4-FAST) using isothermal Gibson assembly. The plasmids allowing the mammalian expression of mTurquoise2-pFAST and EGFP-pFAST tandems under the control of a CMV promoter were constructed by replacing the sequence coding for superYFP or mCherry of the previously described plasmids[35] by pFAST sequence. The plasmid allowing the mammalian expression of mTurquoise2-AURKA-pFAST biosensor under the control of a CMV promoter was constructed by replacing the sequence coding for superYFP in the previously described plasmid by pFAST sequence. All the plasmids used in this study are listed in Supplementary Table 10.

*Cell viability assay.* HeLa cells were treated with MEM media containing the appropriate fluorogen (HBP-3,5DOM, HBP-3,5DM, HBP-3M, HBT-3,5DOM, HBT-3,5DM, HBT-3M, HBO-3,5DM, HBO-3M, HBIR-3,5DOM, HBIR-3,5DM, and HBIR-3M) at the indicated concentrations for the indicated times. The cell viability was evaluated by fluorescence microscopy using the LIVE/DEAD®

viability/cytotoxicity assay kit (Molecular Probes, Life Technologies) following the manufacturer's protocol.

*Fluorescence microscopy.* The confocal micrographs of mammalian cells were acquired on a Zeiss LSM 710 Laser Scanning Microscope equipped with a Plan Apochromat 63 ×/1.4 NA oil immersion objective and on a Leica TCS SP5 confocal laser scanning microscope equipped with a 63 ×/ 1.4 NA oil immersion objective. ZEN and Leica LAS AF softwares were used to collect the data. Zeiss LSM 880 confocal laser scanning microscope equipped with a 63 ×/1.4 NA oil immersion objective and equipped with photomultiplier modules for confocal imaging and a dedicated highly sensitive spectral detector using the Airyscan module was used to acquire improved spatial resolution images of HeLa cells. Airyscan uses a 32-channel gallium arsenide phosphide photomultiplier tube (GaAsP-PMT) area detector that collects a pinhole-plane image at every scan position. Each detector element functions as a single, very small pinhole[60]. ZEN blue and black softwares were used to collect the data and processing of Airyscan images were performed on the software. The images were analyzed with Fiji (Image J).

Photobleaching measurements for HBR-3,5DOM, HBR-3,5DM, HMBR, HBP-3,5DOM, HBP-3,5DM, and HBP-3M were acquired using 405, 458, and 488 nm excitation in different illumination conditions. At 488 nm excitation, EGFP was used as a control. At 458 nm excitation, mTurquoise2 was used as a control. Samples were acquired continuously for 150 images or 500 images using different time lapse between two images. In all cases the pixel dwell was 2.55 µs. The images were analyzed with Fiji (Image J).

*FLIM-FRET.* FLIM analyses were performed with a time-gated custom-built system coupled to a Leica DMI6000 microscope (Leica) with a CSU-X1 spinning disk module (Yokogawa) and a 63 ×/1.4 NA oil immersion objective as described elsewhere[34]. To calculate fluorescence lifetime, five temporal gates with a step of 2 ns each allowed the sequential acquisition of five images covering a total delay time spanning from 0 to 10 ns. The five images were used to calculate the pixel-by-pixel mean fluorescence lifetime as described elsewhere[34,61]. Lifetime measurements and calculations were performed using the Inscoper software (Inscoper). Lifetime was calculated only when pixel-by-pixel fluorescence intensity in the first gate was above 3000 gray levels.

*Super-resolution microscopy.* STimulated Emission Depletion (STED) images were acquired on NeurImag facility with a confocal laser scanning microscope LEICA SP8 STED 3DX equipped with a 93×/1.3 NA glycerol immersion objective and with two hybrid detectors (HyDs). The specimens were imaged with a white-light laser and a pulsed 775 nm depletion laser to acquire nanoscale imaging. Typically, images of 1024 × 1024 pixels were acquired with a magnification above three resulting in a pixel size in the range of 25–45 nm. Deconvolution processing was performed on STED images using CMLE analysis with Huygens software. Iterative processes (up to 40 cycles) were used with a quality criteria ranging from 1 to 5%.

*Flow cytometry analysis.* Flow cytometry on HEK 293 T cells was performed on a Gallios analyzer (Beckman Coulter) equipped with 405 nm, 488 nm, and 638 nm lasers and ten filters and channels. To prepare samples, cells were first grown in cell culture flasks, then transiently co-transfected with pAG104 (cmv-FAST) and pAG753 (cmv-iRFP670) or pAG654 (cmv-pFAST) and pAG753 (cmv-iRFP670) 24 h after seeding, using Genejuice (Merck) according to the manufacturer's protocol for 24 h. Simple positive controls were also prepared by transiently transfected cells with either pAG753 or pAG654 plasmids. After 24 h, cells were centrifuged in PBS with BSA (1 mg/ml) and resuspend in PBS-BSA supplemented with the appropriate amounts of fluorogens. For each experiments, 20,000 cells positively transfected with iRFP670 (Ex 638 nm / Em 660 ± 10 nm) were analyzed with the following parameters: Ex 488 nm, Em 525 ± 20 nm for cells expressing FAST and pFAST labeled with HMBR and HBP-3,5DM; Ex 488 nm, Em 575 ± 15 nm for cells expressing FAST and pFAST labeled with HBR-3,5DM and HBP-3,5DOM and Ex 488 nm, Em 620 ± 15 nm for cells expressing FAST and pFAST labeled with HBR-3,5DOM. Data were analyzed using Kaluza Analysis software (Beckman Coulter).

## Experiments in primary hippocampal neuronal cells.
All experiments involving rats were performed in accordance with the directive 2010/63/EU of the European Parliament and of the Council of 22 September 2010 on the protection of animals used for scientific purposes. Hippocampal neurons from male and female Sprague Dawley embryonic rats (E18) were prepared as described previously[62]. Cells were grown on onto poly-L-lysine-coated 18 mm coverslips (1 mg/ml) at a density of 25,000 cells /cm² in Neurobasal-B27 medium previously conditioned by a confluent glial feeder layer [Neurobasal medium (ThermoFisher 21103049) containing a 2% B27 supplement (ThermoFisher A3582801), and 500 µM L-glutamine (ThermoFisher 25030024)]. Neurons were transfected 2 days before imaging using Lipofectamine 2000 (ThermoFisher). After 10–21 days in vitro, neurons were incubated with the fluorogenic chromophores and imaged immediately at 37 °C.

## Experiments in chicken embryos.
JA57 chicken fertilized eggs were provided by EARL Morizeau (8 rue du Moulin, 28190 Dangers, France) and incubated at 38 °C in a Sanyo MIR-253 incubator. Embryos used in this study were between E2

(HH14) and E3 (HH14 + 24 h). The sex of the embryos was not determined. Under current European Union regulations, experiments on avian embryos between 2 and 4 days in ovo are not subject to restrictions.

*Cloning.* For expression of H2B-FAST and H2B-pFAST in the chick neural tube, the CMV promoters in pAG109 and pAG657 and pAG671 were converted to a CAGGS promoter[63] by replacing the NdeI/BglII fragment with a NdeI/BglII CAGGS fragment from pCAGGS-MCS2 (X. Morin, unpublished). For expression of mito-pFAST, the NdeI/EcoRI fragment of the CMV promoter in pAG671 was replaced with the NdeI/EcoRI CAGGS fragment from pCAGGS-MCS2. pCAGGS-H2B-mCerulean was created by removing an SphI fragment from Tol2-CAG::-Nucbow (Addgene #158992), and pCAGGS-H2B-EYFP was obtained by removing an AgeI fragment from PB-CAG::CytBow (Addgene #158995). Construction details, complete sequences, and plasmids are available upon request.

*Electroporation* in the chick neural tube was performed at embryonic day 2 (E2, HH stage 14), by applying five pulses of 50 ms at 25 V with 100 ms in between, using a square-wave electroporator (Nepa Gene, CUY21SC) and a pair of 5 mm gold-plated electrodes (BTX Genetrode model 512) separated by a 4 mm interval. The DNA solution was injected directly into the lumen of the neural tube via glass capillaries. Bilateral electroporation was achieved by switching the electrodes polarity and repeating the procedure after 2 h. DNA constructs were used at 0.5 µg/µl each, except pCX-mbCherry which was used at a concentration of 0.3 µg/µl. En-face culture of the embryonic neuroepithelium was performed at E3 (24 h after electroporation). After extraction from the egg and removal of extraembryonic membranes in PBS, embryos were transferred to 37 °C F12 medium and pinned down with dissection needles at the level of the hindbrain and hindlimbs in a 35 mm Sylgard dissection dish. A dissection needle was used to separate the neural tube from the somites from hindbrain to caudal end on both sides of the embryo, and the roof-plate was then slit with the needle. The neural tube and notochord were then "peeled off" from the remaining tissues and equilibrated 2 min in 1% low melting point agarose/F12 medium at 38 °C. The tissue was then transferred in a drop of agarose medium to a glass-bottom culture dish (MatTek, P35G-0-14-C) and excess medium was removed so that the neural tube would flatten with its apical surface facing the bottom of the dish, in an inverted open book conformation. After 30 s of polymerization on ice, an extra layer of agarose medium (100 µl) was added to cover the whole tissue and hardened on ice for 1 min. Then 1.9 mL of 37 °C culture medium was added (F12/Penicillin–Streptomycin/Sodium pyruvate) and the culture dish was transferred to the 37 °C chamber of a spinning disk confocal microscope. To image pFAST and FAST, intermediate dilutions (2–200 µM) of ligands HBR-3,5DOM, HBR-3,5DM, HMBR, HBP-3,5DOM, HBP-3,5DM, HBP-3M, and HBIR-3M were prepared by diluting the original 20 mM stocks in F12 medium, and appropriate volumes were added to the dish to reach the desired final concentrations.

*Live imaging* was performed on an inverted microscope (Nikon Ti Eclipse) equipped with a heating enclosure (DigitalPixel, UK), a spinning disk head (Yokogawa CSUW1) and Borealis system (Andor) for confocal imaging, a Spectra-X LED light engine (Lumencor) for widefield fluorescence illumination and an sCMOS Camera (Orca Flash4LT, Hamamatsu) driven by MicroManager software[64]. Image stacks were obtained at 2- or 3-min intervals either with a 10× objective (CFI Plan APO LBDA, NA 0.45, Nikon; z-step = 4 µm) or a 100× oil immersion objective (APO VC, NA 1.4, Nikon; z-step = 1 µm).

**Reporting summary**. Further information on research design is available in the Nature Research Reporting Summary linked to this article.

## Data availability

All data are available in the article and supplementary information. Source data of all data presented in graphs within the figures are provided with this paper. The plasmids used in this study will be available from Addgene. Source data are provided with this paper.

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

## Acknowledgements

We thank K. D. Wittrup, for providing us with the pCTCON2 vector and the EBY100 yeast strain for the yeast display selection. We thank the flow cytometry facility CISA (Cytométrie Imagerie Saint-Antoine) of UMS LUMIC at the Faculty of Medicine of Sorbonne Université, and, more particularly, Annie Munier for her assistance. We thank the imaging facility of the Institut de Biologie Paris Seine of Sorbonne Université. Airyscan and STED experiments were carried out at NeurImag Imaging core facility, part of the IPNP, Inserm 1266 unit and Université de Paris. We thank Leducq establishment for funding the Leica SP8 Confocal/STED 3DX system, Sésame Région Ile-de-France for funding the Zeiss 880 Confocal/Airyscan system, and FLAG-ERA for grant SENSEÏ by ANR-19-HBPR-0003. This work has been supported by the European Research Council (ERC-2016-CoG-724705 FLUOSWITCH), the Agence Nationale de la Recherche (France BioImaging—ANR-10-INBS-04, Morphoscope2 - ANR-11-EQPX-0029, ANR-19-CE13-0026 ADOBE) and a prematuration grant from PSL University and QLife.

## Author contributions

H.B., I.A., G.B., M.T., L.D., N.P., X.M., L.J., and A.G. designed the experiments. H.B., K.O., E.F., R.G., J.N., I.A., T.L.S., A.G.T., C.L., G.B., L.D., N.P., X.M., and A.G. performed the experiments. H.B., E.F., G.B., M.T., L.D., N.P., X.M., L.J., and A.G. analyzed the experiments. H.B. and A.G. wrote the paper with the help of all the authors.

## Competing interests

The authors declare the following competing financial interest: A.G. and L.J. are cofounders and hold equity in Twinkle Bioscience/The Twinkle Factory, a company commercializing the FAST technology. The remaining authors declare no competing interest.
