## [Peer Review File · Nature Communications]

Reviewers' Comments:

Reviewer #1:

Remarks to the Author:

Benaissa et al. report the engineering of a series of chemogenetic reporter molecules for fluorescence imaging applications. Specifically, the authors have developed a series of systematically modified fluorogenic dyes that form brightly fluorescent complexes upon binding to engineered fluorescence-activating proteins. To develop this range of color variants, the authors have elegantly synthesized a series of isosteric chromophore analogues that (by virtue of being isosteric) all fit into the chromophore binding pocket of FAST. In addition, they have employed directed protein evolution, using an elegant system of yeast-display plus FACS, to develop the pFAST protein which exhibits even more promiscuous binding to the chromophores.

This series of dye + protein combinations is analogous to the range of fully genetic fluorescent protein color variants that were developed between ~1995-2005. However, in this substantial and impressive manuscript, the full series of color variants is being reported at one time rather than in a series of papers published over a decade. While there are pros and cons of this system relative to fluorescent proteins, it is my overall opinion that this is a technical tour de force that could be suitable for publication once some issues have been addressed.

There is no getting around the fact that fluorescent proteins represent a very 'high bar' for a new fluorescent reporter technology to have to surpass. Indeed, I would not expect any new technology to surpass fluorescent proteins in all aspects. However, I do feel it is important that a new fluorescent reporter technology should be demonstrated to be able to have 'niche' advantages or to be capable of things that are simply impractical with fluorescent proteins. My main concern is that the chosen demonstrations of the new fluorescent reporters have not demonstrated such a new capability or niche application in which they would have a clear advantage. The one example of a new capability (that is not possible with fluorescent proteins) seems to be switching off fluorescence by adding a dark chromophore. This is interesting but it is not so different from photobleaching of fluorescent proteins to switch off their fluorescence.

The manuscript would be strengthened by a clear demonstration of niche application where the pFAST system has an advantage to fluorescent proteins or enables an application that is impractical with fluorescent proteins. Notably, one half of one paragraph in the Discussion section discusses potential applications of pFAST for FRET. I feel that the advantages suggested in this section are certainly important and they represent 'features' of pFAST that could not be easily replicated with fluorescent proteins. Unfortunately, the authors did not provide any proof-of-concept data to support the potential application of pFAST for FRET applications. Doing so would help to provide a convincing case for the utility of these new chemogenetic reporter molecules.

Major concerns:

It was confusing that the directed evolution work is presented as an effort to generate a promiscuous variant, since the most obvious way to do this would have been to use different chromophores in different rounds of selection. Since only one chromophore was used for each selection, it certainly seems as though the original goal of this effort must have actually been to generate specific binders for each chromophore (which is more logical and would be more useful for multicolour imaging applications). I could be mistaken, but I expect that the promiscuous binding was not the intended outcome. If this is the case, I feel that the text could be reworded a bit to make it clear about the actual goal of the work. I don't feel that presenting this as an unexpected negative outcome will detract from the overall quality and impact of the work.

On page 6 it is stated that pFAST shows proper cellular localization in fixed cells. However, it is apparent from the images in Fig. S12 that none of the fluorescent labels tested (HaloTag, Venus, and pFAST) show proper cellular localization in the fixed cells (assuming that the observed localization in the live cell is proper). I don't feel that the conclusion is justified based on this data.

In Figure 6, most examples show a comparison between confocal images and deconvolve STED images. For this to be a fair comparison, both confocal and STED images would need to be

processed using the same procedures. Ideally, no extra deconvolution processing should be used for any of the images.

Since GFP is considered to be only just photostable enough for use in STED (or so I have heard), it would be valuable for the authors to include an analysis of photobleaching comparison of pFAST:HBR3,5DOM vs GFP during STED. Improved photostability would be an important advantage that would provide a convincing justification for using pFAST for live cell STED imaging.

Minor comments:

Introduction: should specify that "permeant" refers to "membrane permeant".

I found it a bit confusing to refer to various series of chromophores as X = P, etc, rather than just writing HBP, etc., which is fewer characters (including spaces) and easier to understand.

For figure 2h, the figure legend will require an explanation of what circle size, empty/filled, and left/right coloring, and top/bottom coloring, all mean.

The phrase 'tridimensional crystal structure' is a bit strange. I feel it is clear that a crystal structure must be three dimensional.

Page 5: "did not alter significantly fluorescence" would be better written as "did not significantly alter fluorescence"

Page 7: The statement that HBIR-3M served to "efficiently reverse the labeling of pFAST " is a bit misleading. The pFAST is still labeled, just with a non-fluorescent chromophore.

The rationale for choosing the 3,5DM chromophores (as opposed to other chromophores) for the directed evolution should be stated.

Reviewer #2:

Remarks to the Author:

Arnaud Gautier et al. in the submitted manuscript "Engineering of a fluorescent chemogenetic reporter with tunable color for advanced live-cell imaging" report improvements of the Fluorescence-Activating and absorption-Shifting Tag (FAST) and synthesis of a series of hydroxybenzylidene rhodanine (HBR) derivatives followed by extensive characterization fusing on microscopy applications. The combinatorial library of variants generated by random mutagenesis in combination with yeast display allowed identification of three improved variants: oFAST, tFAST and pFAST. The best performing variants was named "promiscuous FAST" or pFAST. The HBR derivatives are produced using standard synthetic routes with a good yields. The pFAST in combination with HBR derivatives are characterized by measuring photophysical properties, cytotoxicity, cell permeability and binding unbinding kinetics. Importantly, the authors demonstrate extensive applications for imaging of tagged proteins in living cells and organisms. In general, pFAST should be applicable to multiple biological studies.

Overall, the manuscript is interesting and represents considerable improvement of FAST method, but lacks some important experiments and corrections. Thus I happy to recommend this manuscript to be accepted for publishing in Nature Communications after a major revision:

Major points:

1. Authors are comparing photostability of pFAST with EGFP, but they use synthetic fluorogenic which need to be added externally to the cells before imaging. Thus it would be much more adequate to compare pFAST with tagging methods utilizing synthetic cell-permeable fluorophores.
2. In the figure 6 only profiles of confocal, STED and deconvolved STED is provided. No quantification of the resolution improvement is performed. For readers it is important to present quantification because the obtained number could be compared with other labeling methods directly. Please include this piece of data.

Minor points:

1. In the introduction authors mention that the reported pFAST is able to span "entire visible spectrum" which is 400-700 nm, but the most red shifted fluorogen HBIR-3.5DOM has emission maximum at 616 nm. This corresponds to the orange color (not red as stated by authors). Please correct these misleading statements in the manuscript text.
2. In the figure 4c no color calibration bar of z-scale is provided. Please include it. The authors highlight in the discussion that 3D STED nanoscopy was performed, but no experimental data are provided. The figure 6 presents only 2D STED image. Please correct this.
3. The most of the imaging experiments utilize 488 nm excitation laser (as indicated in Table S13) which is quite phototoxic for living cell and organism. The authors should indicate this and propose improvements in the discussion part of the manuscript.
4. FAST-HBR system displays considerable stokes shift. Is it possible to perform two-color or three-color imaging experiment which uses single excitation laser source (for example 488 nm) in combination with multiple detection windows? Such application would be attractive because single exposure to the excitation light could be enough to obtain image of multiple structures. The authors should discuss such possibility or even perform the experiment.
5. Please provide quantification of two-color fluorescence viability assay. It is not possible to see the effect from the provided images in Figure S7.
6. The molecular modeling experiment is very interesting and useful for understanding pFAST properties, but it is still just a model which need to be verified. Have authors tried to perform reverse mutation to see if the proposed function is indeed matching the experimental data? Please comment this in the manuscript.
7. In the Figure S14 some curves are not reaching plateau at high HMR or HBP concentrations. This indicates that labelling of the tag protein is not fully reached. In some cases FAST appears brighter compared to pFAST. Authors should discuss this in more details in the manuscript text.

Reviewer #3:

Remarks to the Author:

This manuscript details an effort to develop a variant of FAST protein designed to bind a range of chromophores. The protein that results from these efforts, pFAST is able to bind a range of chromophores with useful binding constants. The basis for this binding is examined via modeling. The resulting imaging demonstrates several applications, including several that were possible with previous FAST proteins. However, the potential benefit of binding of being able used multiple partners is clear. Of the issues raised below, the potential for oxygen sensitization should be addressed in some fashion.

1. One significant issue is that is not addressed in any fashion is the role of singlet oxygen sensitization in this strategy. Most of the red-shifted molecule incorporate sulfur atoms, which are well known to promote intersystem crossing. This issue is not just a curiosity, since the authors propose to use in live organisms where phototoxicity may be a significant issue. To address this, the authors should compare the in vitro singlet oxygen generation of the proteins (which can be done using commercial singlet oxygen sensors) and the phototoxicity of these constructs in cells.
2. The magnitude of the protein binding induced turn-ON effect is not characterized in detail. This may differ among this panel of compounds. The effect should be examined by looking the optical properties of the free compounds in water and viscous organic solvent (e.g. ethylene glycol) and also by imaging wild-type (non-transfected) cells with these compounds.
3. The exact benefits for this pFAST approach relative to existing technologies are only addressed in the initial characterization and some initial imaging comparison. The only application that is clearly unique to FAST approach is the turn-off work. However, this type of loss of signal could also be achieved by simply bleaching a conventional self-labeling fluorophore-protein construct. It would be useful to have some comparison data comparing pFAST to conventional FPs (at least) in a complex application. For example, for the STED imaging some comparison to conventional FP strategy should be included.

4. The synthesis and characterization of the small molecules could be improved. The current field standard is to include NMR spectra. Also, some synthetic scheme should be included to guide the reader.

Response to reviewers

Reviewer #1

Benaissa et al. report the engineering of a series of chemogenetic reporter molecules for fluorescence imaging applications. Specifically, the authors have developed a series of systematically modified fluorogenic dyes that form brightly fluorescent complexes upon binding to engineered fluorescence-activating proteins. To develop this range of color variants, the authors have elegantly synthesized a series of isosteric chromophore analogues that (by virtue of being isosteric) all fit into the chromophore binding pocket of FAST. In addition, they have employed directed protein evolution, using an elegant system of yeast-display plus FACS, to develop the pFAST protein which exhibits even more promiscuous binding to the chromophores.

This series of dye + protein combinations is analogous to the range of fully genetic fluorescent protein color variants that were developed between ~1995-2005. However, in this substantial and impressive manuscript, the full series of color variants is being reported at one time rather than in a series of papers published over a decade. While there are pros and cons of this system relative to fluorescent proteins, it is my overall opinion that this is a technical tour de force that could be suitable for publication once some issues have been addressed.

There is no getting around the fact that fluorescent proteins represent a very 'high bar' for a new fluorescent reporter technology to have to surpass. Indeed, I would not expect any new technology to surpass fluorescent proteins in all aspects. However, I do feel it is important that a new fluorescent reporter technology should be demonstrated to be able to have 'niche' advantages or to be capable of things that are simply impractical with fluorescent proteins. My main concern is that the chosen demonstrations of the new fluorescent reporters have not demonstrated such a new capability or niche application in which they would have a clear advantage. The one example of a new capability (that is not possible with fluorescent proteins) seems to be switching off fluorescence by adding a dark chromophore. This is interesting but it is not so different from photobleaching of fluorescent proteins to switch off their fluorescence.

1.1. We thank reviewer 1 very much for acknowledging the quality and relevance of our study. We fully agree that a new technology should be capable of doing things that are not possible to do with classical FPs. FAST was previously shown to have several advantages over fluorescent proteins such as being fluorescent on-demand, be instantaneously fluorescent, and be fully functional in anaerobic organisms such as *Clostridium*. In this study, we further expand the possibilities offered by this technology by engineering a promiscuous variant (pFAST) capable of binding fluorogens with various spectral properties, and having globally enhanced properties over FAST.

One unique application we showed in the original manuscript was indeed the ability to switch off very rapidly the fluorescence of pFAST in cells and in embryo tissues by addition of a high-affinity dark competitor. Although, this can appear not so different from photobleaching a FP, we believe that it provides some important advantages. In

particular it does not require high light intensities, which can be toxic for the specimen, and can be easily done on a very large tissue which would be very challenging to do by photobleaching. In addition to this first demonstration, we show now in the revised manuscript that the dark competitor can be itself replaced by addition of an excess of fluorogenic chromophore, enabling to switch fluorescence on again (See Figure S18j-l in the new version). Such experiments would not be possible with photobleaching. The text of the main text has been modified accordingly.

The manuscript would be strengthened by a clear demonstration of niche application where the pFAST system has an advantage to fluorescent proteins or enables an application that is impractical with fluorescent proteins. Notably, one half of one paragraph in the Discussion section discusses potential applications of pFAST for FRET. I feel that the advantages suggested in this section are certainly important and they represent 'features' of pFAST that could not be easily replicated with fluorescent proteins. Unfortunately, the authors did not provide any proof-of-concept data to support the potential application of pFAST for FRET applications. Doing so would help to provide a convincing case for the utility of these new chemogenetic reporter molecules.

1.2. We now present in the revised manuscript a fully new set of experiments showing that pFAST displays unprecedented features for optimizing the design of FRET biosensors. FRET biosensors have become essential tools for studying signal transduction pathways in cells. Their design and optimization remains challenging and labor-intensive because (i) various FRET pairs must be tested to obtain the best FRET signal, and (ii) proper characterization of FRET efficiency requires acceptor-free constructs to measure the loss of fluorescence intensity (or the change of lifetime) undergone by a donor when in close proximity of an acceptor. We demonstrated that the FRET efficiency in biosensors containing pFAST as acceptor could be (i) easily optimized by testing various chromophores, and (ii) directly quantified in a single experiment by measuring the fluorescence loss or lifetime variation upon addition of the fluorogen (**Supplementary Fig. 20** and **Figure 6** in the new manuscript). We showed moreover that the possibility to test various acceptors in the exact same cellular context through sequential labeling allowed for optimizing FRET biosensors at the single cell level. This set of experiments enabled us moreover to show that the ability to form dark complexes absorbing in the green region opened interesting prospects for the design of dark acceptors for FRET measurements based on fluorescence lifetime imaging microscopy (FLIM) ; indeed as pFAST labeled with a dark chromophore does not fluoresce, it can thus modulate the fluorescence lifetime of a donor without blocking an imaging channel. Overall, this set of experiments demonstrates that the tunability and versatility of pFAST provides unprecedented means to quantify and optimize the efficiency of FRET biosensors in live cells. A paragraph has been added in the *Results* part to present these new data, and the *Discussion* has been modified accordingly.

Major concerns:

It was confusing that the directed evolution work is presented as an effort to generate a promiscuous variant, since the most obvious way to do this would have been to use

different chromophores in different rounds of selection. Since only one chromophore was used for each selection, it certainly seems as though the original goal of this effort must have actually been to generate specific binders for each chromophore (which is more logical and would be more useful for multicolour imaging applications). I could be mistaken, but I expect that the promiscuous binding was not the intended outcome. If this is the case, I feel that the text could be reworded a bit to make it clear about the actual goal of the work. I don't feel that presenting this as an unexpected negative outcome will detract from the overall quality and impact of the work.

1.3 Our objective was to generate a universal protein tag that could bind and stabilize the fluorescent state of fluorogenic chromophores displaying various spectral properties to simplify biologists' work. As we show in this paper, the chromophore promiscuity of pFAST provides many imaging possibilities with a single tag, simplifying experimental workflow. As the way we present the directed evolution experiment reflects our original goal, we kept the original text.

On page 6 it is stated that pFAST shows proper cellular localization in fixed cells. However, it is apparent from the images in Fig. S12 that none of the fluorescent labels tested (HaloTag, Venus, and pFAST) show proper cellular localization in the fixed cells (assuming that the observed localization in the live cell is proper). I don't feel that the conclusion is justified based on this data.

1.4 We show on **Figure 4b** that pFAST fusions show the same localizations both in live cells and in fixed cells, except in the case of MAP4-pFAST that shows improper localization in fixed cells, as originally indicated in the legend of **Figure 4**. Similar mislocalization was observed upon fixation of MAP4-HaloTag and MAP4-Venus (**Supplementary Fig. 12**) suggesting that the mislocalization of MAP4-pFAST fixed cells was an artefact induced by fixation. We modified the text on page 6 to make this point clearer.

In Figure 6, most examples show a comparison between confocal images and deconvolve STED images. For this to be a fair comparison, both confocal and STED images would need to be processed using the same procedures. Ideally, no extra deconvolution processing should be used for any of the images.

1.5 Deconvolution is commonly done in STED microscopy in order to increase the contrast of STED images and improve their rendering without affecting the resolution. STED deconvolution by the Huygens software uses a measured or computed STED PSF to deconvolve. Deconvolution mainly increases the contrast of STED images (which have a low signal-to-noise ratio because few photons are collected), and has a limited effect on resolution (see Schoonderwoert et al., *Microscopy Today*, 2013 doi:10.1017/S1551929513001089). We showed in the original version of the paper the comparison of confocal, untreated STED and deconvolved STED images on one example showing plasma membrane details (Fig 6a,b,c in the old version or Fig 7a,b,c in the revised manuscript). The untreated STED image convincingly showed increased resolution over the confocal image, while deconvolution mainly increased contrast and

had a limited effect on resolution. In the revised manuscript, we now show also side-by-side comparisons of confocal, untreated STED and deconvolved STED images of microtubules in HeLa cells (See **Figure 7e-k**). As suggested by the reviewer, and to be totally transparent, we also added in Supplementary information on **Supplementary Fig. 22** fifteen examples where we systematically show untreated and deconvolved confocal images together with untreated and deconvolved STED images allowing one to fully evaluate the contribution of deconvolution. We showed as above that deconvolution has a limited effect on resolution and that it mainly improves contrast and smoothing, whereas resolution is mainly improved by STED imaging.

Since GFP is considered to be only just photostable enough for use in STED (or so I have heard), it would be valuable for the authors to include an analysis of photobleaching comparison of pFAST:HBR3,5DOM vs GFP during STED. Improved photostability would be an important advantage that would provide a convincing justification for using pFAST for live cell STED imaging.

1.6 Direct comparison of the performance of conventional EGFP and pFAST:HBR3,5DOM during STED imaging would be indeed interesting. However, we reasoned that side-by-side comparison would involve two different depletion lasers (595 nm for EGFP and 775 nm for pFAST:HBR3,5DOM) that would probably favor pFAST because the 775 nm pulsed laser is known to be more effective. To be completely fair in our comparison and given that the use of 775 nm STED laser is preferred for live-cell STED imaging as light is less toxic for live specimens at this wavelength, we decided to compare pFAST with conventional fluorescent protein, that would respond under the same 775 nm depletion laser. So we now present in the revised manuscript a comparison with mCherry and HaloTag labeled with the tetramethylrhodamine dye, which both have spectral properties comparable with pFAST:HBR-3,5DOM (**Supplementary Fig. 23**). Whereas the three probes displayed comparable photostability in confocal microscopy (**Supplementary Fig. 10d** and **Supplementary Movies 9-11**), we observed that mCherry photobleached very rapidly under STED conditions and did not allow us to generate convincing STED images (**Supplementary Fig. 23**). On the other hand, pFAST:HBR-3,5DOM and HaloTag-TMR showed higher resistance to photobleaching under STED conditions, and enabled us to obtain STED images with comparable qualities (**Supplementary Fig. 23**). The main text has been modified accordingly.

Minor comments:

Introduction: should specify that “permeant” refers to “membrane permeant”.

1.7 The text has been corrected accordingly in the revised manuscript.

I found it a bit confusing to refer to various series of chromophores as X = P, etc, rather than just writing HBP, etc., which is fewer characters (including spaces) and easier to understand.

1.8 The text has been corrected in the revised manuscript to avoid any confusion.

For figure 2h, the figure legend will require an explanation of what circle size, empty/filled, and left/right coloring, and top/bottom coloring, all mean.

1.9 The circle diameter reflects the value of the fluorescence quantum yield, while the different coloring systems are used to tell apart the different variants (FAST, oFAST, tFAST and pFAST). We modified the legend accordingly.

The phrase 'tridimensional crystal structure' is a bit strange. I feel it is clear that a crystal structure must be three dimensional.

1.10 The phrase 'tridimensional crystal structure' has been replaced by 'crystal structure'

Page 5: "did not alter significantly fluorescence" would be better written as "did not significantly alter fluorescence"

1.11 We thank reviewer 1 for this suggestion and have changed the sentence accordingly.

Page 7: The statement that HBIR-3M served to "efficiently reverse the labeling of pFAST" is a bit misleading. The pFAST is still labeled, just with a non-fluorescent chromophore.

1.12 We thank reviewer 1 for this remark and have modified the text to specify that HBIR-3M can serve to switch off the fluorescence of pFAST through replacement with a dark chromophore.

The rationale for choosing the 3,5DM chromophores (as opposed to other chromophores) for the directed evolution should be stated.

1.13 There is no rationale for choosing 3,5DM chromophores. We actually did selection with 3M and 3,5DOM chromophores as described in Supplementary Text 1 and Fig S2 of the original manuscript.

Reviewer #2

Arnaud Gautier et al. in the submitted manuscript "Engineering of a fluorescent chemogenetic reporter with tunable color for advanced live-cell imaging" report improvements of the Fluorescence-Activating and absorption-Shifting Tag (FAST) and synthesis of a series of hydroxybenzylidene rhodanine (HBR) derivatives followed by extensive characterization focusing on microscopy applications. The combinatorial library of variants generated by random mutagenesis in combination with yeast display allowed identification of three improved variants: oFAST, tFAST and pFAST. The best performing variant was named "promiscuous FAST" or pFAST. The HBR derivatives

are produced using standard synthetic routes with a good yields. The pFAST in combination with HBR derivatives are characterized by measuring photophysical properties, cytotoxicity, cell permeability and binding unbinding kinetics. Importantly, the authors demonstrate extensive applications for imaging of tagged proteins in living cells and organisms. In general, pFAST should be applicable to multiple biological studies.

Overall, the manuscript is interesting and represents considerable improvement of FAST method, but lacks some important experiments and corrections. Thus I happy to recommend this manuscript to be accepted for publishing in Nature Communications after a major revision:

2.1. We thank reviewer 2 very much for acknowledging the interest of our study for biological studies.

Major points:

1. Authors are comparing photostability of pFAST with EGFP, but they use synthetic fluorogenic which need to be added externally to the cells before imaging. Thus it would be much more adequate to compare pFAST with tagging methods utilizing synthetic cell-permeable fluorophores.

2.2. EGFP being the most used fluorescent probe in cell biology, we believe that comparing the photostability of pFAST with EGFP is a fair comparison. It allows cell biologists interested in using this new technology to have a comparison with a probe they know and use. To address nevertheless this comment, we compared pFAST to mCherry and HaloTag labeled with commercially available tetramethylrhodamine (see comment **# 1.6**) in STED imaging. We also added in the revised manuscript on **Supplementary Figure 10d** additional photostability data showing a comparison of pFAST:HBR-3,5DOM with the red fluorescent protein mCherry and HaloTag labeled with commercially available tetramethylrhodamine.

2. In the figure 6 only profiles of confocal, STED and deconvolved STED is provided. No quantification of the resolution improvement is performed. For readers it is important to present quantification because the obtained number could be compared with other labeling methods directly. Please include this piece of data.

2.3. To properly quantify the resolution improvement is not a trivial task as it necessitates the labeling of objects of known size. Moreover, the comparison to other labeling methods would require to image the same structure labeled by different techniques using the same microscopy settings. To do so, nuclear pores have been elegantly used recently as reference standards for comparing super-resolution techniques and labeling methods (Nat. Methods 2019 10.1038/s41592-019-0574-9). Such extensive study was however according to us out-of-scope of the current work. That is why we chose in the original version of the paper to present intensity profiles that, albeit imperfect, are widely and commonly used to demonstrate and quantify the

gain in resolution in super-resolution imaging. We now provide in the revised manuscript additional data showing the resolution improvement we observed when imaging microtubules labeled with MAP4-pFAST (**Supplementary Fig. 22**). Quantification was performed on fifteen STED images. We also present on **Fig. 7** additional zoom inserts enabling to better appreciate the gain in resolution. We believe that this new set of data enables to appreciate the gain of resolution and to assess the performance of our probe for STED imaging.

Minor points:

1. In the introduction authors mention that the reported pFAST is able to span “entire visible spectrum” which is 400-700 nm, but the most red shifted fluorogen HBIR-3.5DOM has emission maximum at 616 nm. This corresponds to the orange color (not red as stated by authors). Please correct these misleading statements in the manuscript text.

2.4. The definition of what is a ‘red fluorescent’ probe is often a matter of debate. DsRed, the first ‘red’ fluorescent protein, has an emission maximum at 588 nm, while mCherry, one of the most used ‘red’ fluorescent protein, has an emission maximum at 615 nm. Vladislav Verkhusha proposed in the review ‘Guide to Red Fluorescent Proteins and Biosensors for Flow Cytometry’ in *Methods in Cell Biology* (2011) 102, 431-461 to divide all red fluorescent proteins into three categories according to their fluorescence emission maximum: orange (with emission from 550 to 590 nm), red (with emission maximum from 590 to 630 nm) and far-red (with emission maximum more than 630 nm). According to these definitions, we believe it is correct to define the fluorescence of pFAST:HBIR-3,5DOM as red in order to allow a direct comparison with fluorescent proteins. We modified the introduction to mention that pFAST is able to span “the visible spectrum from blue (λ_{em} 473 nm) to red (λ_{em} 616 nm)” rather than “the entire visible spectrum”

2. In the figure 4c no color calibration bar of z-scale is provided. Please include it. The authors highlight in the discussion that 3D STED nanoscopy was performed, but no experimental data are provided. The figure 6 presents only 2D STED image. Please correct this.

2.5. Color calibration bar for the z-scale has been added on figure 4c. About the remark about the 3D STED nanoscopy, we rephrase the text because our previous version was apparently misleading. We did not perform 3D imaging with STED, but we used 3D-STED technology for imaging, which is quite different. By opposition to 2D-STED, in which a doughnut-shaped focal STED intensity distribution is used to enhance the resolution in the lateral plane of the sample, 3D-STED uses a 3D depletion system in order to enhance both the axial and lateral resolutions leading to a more spherical PSF. Although the images shown in figures are 2D images, they were acquired using 3D-STED technology. The text has been modified to precise this subtlety.

3. The most of the imaging experiments utilize 488 nm excitation laser (as indicated in Table S13) which is quite phototoxic for living cell and organism. The authors should indicate this and propose improvements in the discussion part of the manuscript.

2.6. Although the phototoxicity of the 488 nm excitation is a general issue in biological imaging, the 488 nm excitation remains widely used in biological imaging. One of the reasons is that the probes for other laser lines do not reach the performance of the probes for 488 nm excitation. We precise now in the discussion that “It should be noted that the spectral properties are however not yet fully optimal as the blue and green lights used for excitation can be toxic for cells and organisms” and that “future engineering efforts will focus on further red-shifting the absorption and emission properties in order to allow the use of less toxic red and far-red excitation lights for more biocompatible imaging as initiated recently with the development of far-red(fr)FAST that uses fluorogens with an elongated π system.”.

4. FAST-HBR system displays considerable stokes shift. Is it possible to perform two-color or three-color imaging experiment which uses single excitation laser source (for example 488 nm) in combination with multiple detection windows? Such application would be attractive because single exposure to the excitation light could would be enough to obtain image of multiple structures. The authors should discuss such possibility or even perform the experiment.

2.7. As suggested by reviewer 2, it is indeed possible to perform two-color imaging using a single excitation because of the high Stokes shift of our systems. We included experiments in the revised version showing our ability to image pFAST:HBR-3,5DOM together with EGFP using a single 488 nm excitation in chicken embryo tissue (**Supplementary Fig. 17**). We demonstrate moreover that pFAST:HBR-3,5DOM outperformed the long-Stokes-shift fluorescent protein CyOFP1. The main text has been modified to include the description of this new set of experiments.

5. Please provide quantification of two-color fluorescence viability assay. It is not possible to see the effect from the provided images in Figure S7.

2.8. Quantification has been added in the revised version.

6. The molecular modeling experiment is very interesting and useful for understanding pFAST properties, but it is still just a model which need to be verified. Have authors tried to perform reverse mutation to see if the proposed function is indeed matching the experimental data? Please comment this in the manuscript.

2.9. Reverse engineering could be indeed very interesting to verify the role of each mutations. As pFAST contains 10 mutations relative to prototypical FAST, such experiments were however out of the scope of this study, and were not performed. We added sentences in the discussion to describe the results of our modeling studies and

indicate that deeper reverse engineering studies could be performed to confirm the role of each mutation.

7. In the Figure S14 some curves are not reaching plateau at high HMR or HBP concentrations. This indicates that labelling of the tag protein is not fully reached. In some cases FAST appears brighter compared to pFAST. Authors should discuss this in more details in the manuscript text.

2.10. The experiments of **Supplementary Fig. 14** are meant to show the labeling efficiency of FAST and pFAST in cells with different fluorogens. As mentioned in the initial version of the manuscript “the labeling efficiency in live HeLa cells by flow cytometry showed that full labeling of pFAST was achieved at lower chromophore concentrations than FAST, in agreement with its superior binding affinity”. In all fluorogen tested (except for HBP-3,5DOM) pFAST reach saturation at 10 μ M fluorogen, while FAST only reach saturation with HBR-3,5DM and HMBR. The difference of fluorescence intensity observed between FAST and pFAST can be due to differences in brightness or expression levels.

Reviewer #3 (Remarks to the Author):

This manuscript details an effort to develop a variant of FAST protein designed to bind a range of chromophores. The protein that results from these efforts, pFAST is able to bind a range of chromophores with useful binding constants. The basis for this binding is examined via modeling. The resulting imaging demonstrates several applications, including several that were possible with previous FAST proteins. However, the potential benefit of binding of being able used multiple partners is clear. Of the issues raised below, the potential for oxygen sensitization should be addressed in some fashion.

1. One significant issue is that is not addressed in any fashion is the role of singlet oxygen sensitization in this strategy. Most of the red-shifted molecule incorporate sulfur atoms, which are well known to promote intersystem crossing. This issue is not just a curiosity, since the authors propose to use in live organisms where phototoxicity may be a significant issue. To address this, the authors should compare the in vitro singlet oxygen generation of the proteins (which can be done using commercial singlet oxygen sensors) and the phototoxicity of these constructs in cells.

3.1. We thank reviewer 3 for this relevant remark. The question of the phototoxicity is indeed crucial for live-cell imaging. Almost all the experiments shown in the paper have been performed in live cells or live tissues, demonstrating according to us the compatibility of our system with live-cell imaging. In particular, we showed that pFAST could be used to study dynamic processes in chicken embryo tissues, and to image proteins in delicate cells such as dissociated hippocampal neurons. We observed no apparent phototoxicity during the different experiments we performed. A full

characterization of the phototoxicity implies to study the production of singlet oxygen but also of other reactive oxygen species that can be involved in phototoxicity. Such study would be indeed very interesting but deserve according to us a full and independent study. We believe that this study is out of the scope of this paper.

2. The magnitude of the protein binding induced turn-ON effect is not characterized in detail. This may differ among this panel of compounds. The effect should be examined by looking the optical properties of the free compounds in water and viscous organic solvent (e.g. ethylene glycol) and also by imaging wild-type (non-transfected) cells with these compounds.

3.2. The characterization of the turn-ON effect was addressed in the original manuscript. We provided the fluorescence spectra of all fluorogen without and with pFAST on **Supplementary Fig. 5**, allowing to appreciate the turn-ON effect, and we quantified the signal-to-background ratio in cells on **Supplementary Fig. 8**. To complement these data, we added in the new version of the manuscript the fluorescence fold increase undergone by each fluorogen on **Supplementary Fig. 5**. Most fluorogens show turn-on effect > 100.

3. The exact benefits for this pFAST approach relative to existing technologies are only addressed in the initial characterization and some initial imaging comparison. The only application that is clearly unique to FAST approach is the turn-off work. However, this type of loss of signal could also be achieved by simply bleaching a conventional self-labeling fluorophore-protein construct.

3.3. See answer **#1.1**.

It would be useful to have some comparison data comparing pFAST to conventional FPs (at least) in a complex application. For example, for the STED imaging some comparison to conventional FP strategy should be included.

3.4. Systematic side-by-side comparison of pFAST with various fluorescent proteins in single chicken embryos was actually presented in the first version of the manuscript (see Fig 5a,g). This experiment enabled us to provide a side-by-side comparison of the fluorescent probes in the very same embryo, enabling to compare the performances of the probes in the very same cellular context and with the exact same imaging settings. We believe that this experiment provides already some solid and convincing data to compare pFAST to conventional FPs in complex systems.

In addition, as mentioned in **#1.6.**, we now present in the revised manuscript a comparison with mCherry and HaloTag labeled with the tetramethylrhodamine dye, which both have spectral properties comparable with pFAST:HBR-3,5DOM (**Supplementary Fig. 23**). Whereas the three probes displayed comparable photostability in confocal microscopy (**Supplementary Fig. 10d** and **Supplementary Movies 9-11**), we observed that mCherry photobleached very rapidly under STED conditions and did not allow us to generate convincing STED images (**Supplementary**

Fig. 23). On the other hand, pFAST:HBR-3,5DOM and HaloTag-TMR showed higher resistance to photobleaching under STED conditions, and enabled us to obtain STED images with comparable qualities (**Supplementary Fig. 23**). The main text has been modified accordingly.

4. The synthesis and characterization of the small molecules could be improved. The current field standard is to include NMR spectra. Also, some synthetic scheme should be included to guide the reader.

3.5. Synthetic schemes have been added to guide the reader (see **Supplementary Fig. 24**). We believe that the characterization of the small molecules is now consistent with the guideline of Nature Communications (<https://www.nature.com/ncomms/submit/chemical-characterisation>).

Reviewers' Comments:

Reviewer #1:

Remarks to the Author:

The authors have thoroughly addressed my concerns. I recommend the manuscript be accepted in its current form.

Reviewer #2:

Remarks to the Author:

Arnaud Gautier et al. in the re-submitted manuscript "Engineering of a fluorescent chemogenetic reporter with tunable color for advanced live-cell imaging" have addressed most of the concerns. Thus, I am happy to recommend this manuscript to be accepted for publishing in Nature Communications.

Minor remark: It is more understandable for readers if resolution improvement in STED images is supported by numbers and statistical analysis of multiple profiles, show in Figure 7 and supplementary Figure 22. I would strongly suggest making such effort which would emphasize the potential of FAST tags.

Reviewer #3:

Remarks to the Author:

The revised manuscript is significantly improved, and suitable for publication.

Response to reviewers

Reviewer #1 (Remarks to the Author):

The authors have thoroughly addressed my concerns. I recommend the manuscript be accepted in its current form.

We thank reviewer 1 for recommending the publication of this paper in Nature Communications.

Reviewer #2 (Remarks to the Author):

Arnaud Gautier et al. in the re-submitted manuscript “Engineering of a fluorescent chemogenetic reporter with tunable color for advanced live-cell imaging” have addressed most of the concerns. Thus, I am happy to recommend this manuscript to be accepted for publishing in Nature Communications.

We thank reviewer 2 for recommending the publication of this paper in Nature Communications.

Minor remark: It is more understandable for readers if resolution improvement in STED images is supported by numbers and statistical analysis of multiple profiles, show in Figure 7 and supplementary Figure 22. I would strongly suggest making such effort which would emphasize the potential of FAST tags.

As mentioned in our previous response to reviewers, we believe that presenting intensity profiles (that are widely and commonly used to demonstrate improvement in resolution in super-resolution imaging) is the most appropriate way to present the resolution improvement in our STED images. We present many images of various cellular structures having different size and geometry rendering difficult the comparison and statistical analysis of multiple profiles. As we mentioned in our previous response to reviewers (comment #2.3) for our revised manuscript, statistical analysis of the resolution improvement would necessitate the labeling of reference objects of known size, such as nuclear pores. We believe that such extensive study was out-of-scope of the current work. The goal of our STED experiments was to demonstrate that our system was suitable for STED in live cells and neurons. We believe that the multiple experiments we present in this paper demonstrate this point, and that the way we present the performance of our probes for STED imaging is comparable to other studies recently published in the field (see for instance: A general strategy to develop cell permeable and fluorogenic probes for multicolour nanoscopy, Wang, Tran, D’Este, Roberti, Kock, Xue Johnsson, Nat. Chem. **12**, 165–172 (2020)).

Reviewer #3 (Remarks to the Author):

The revised manuscript is significantly improved, and suitable for publication.

We thank reviewer 3 for acknowledging the improvement of our paper and its suitability for publication in Nature Communications.